# GNSS Radio Occultation Soundings from Commercial Off-the-Shelf Receivers Onboard Balloon Platforms

Kevin J. Nelson[1,*], Feiqin Xie[1], Bryan C. Chan[2], Ashish Goel[2], Jonathan Kosh[2], Tyler G. R. Reid[2], Corey R. Snyder[2], and Paul M. Tarantino[2]

[1]Texas A&M University - Corpus Christi, Corpus Christi, TX, USA
[2]Night Crew Labs, LLC, Woodside, CA, USA
[*]Author now at Jet Propulsion Laboratory, California Institute of Technology, Pasadena, CA, USA

**Correspondence:** Kevin J. Nelson (kevin.j.nelson@jpl.nasa.gov)

**Abstract.** The global Navigation Satellite System (GNSS) radio occultation (RO) technique has proven to be an effective tool for Earth atmosphere profiling. Traditional spaceborne RO satellite constellations are expensive with relatively low sampling density for specific regions of interest. In contrast, airborne RO platforms can provide much higher spatial and temporal sampling of ROs around regional weather events. This study explores the capability of a low-cost and scalable Commercial-Off-The-Shelf (COTS) GNSS receiver onboard high-altitude balloons. The refractivity retrievals from balloon-borne RO payloads obtained from two flight campaigns (World View and ZPM-1) are presented. The balloon-borne RO soundings from the World View campaign show refractivity profiles between 6 and 19 km with overall near-zero median difference from colocated ECMWF ERA5 reanalysis data and variability comparable to spaceborne RO missions (~2.3% median-absolute-deviation). Soundings from the ZPM-1 campaign show a relatively large positive refractivity bias (~2.5%). In summary, low-cost COTS RO payloads onboard balloon platforms are worth further engineering and study in order to provide capabilities for dense, targeted atmospheric soundings that can improve regional weather forecasts via data assimilation.

## 1 Introduction

The radio occultation (RO) atmospheric profiling method was first developed to measure planetary atmospheres in our solar system. The first major use of RO was part of the mission to examine atmospheric profiles of Venus as part of the Mariner V mission launched in 1967 (Fjeldbo and Eshleman, 1969; Fjeldbo et al., 1971). Due to vertical atmospheric density gradients, radio signals transmitted from the spacecraft will bend and be slightly delayed when passing through the limb of the planetary atmosphere before arriving at a receiving antenna on Earth. This bending accumulation along the ray path can be precisely measured using excess phase, and can be used to derive atmospheric pressure, temperature, and concentration of atmospheric constituents (e.g., sulfuric acid concentrations on Venus). The same method was later applied to Mars (Mariner IX, Kliore et al., 1972; Lindal et al., 1979) and Neptune (Voyager II, Lindal, 1992). Even as recently as 2017, additional RO missions to Venus were underway (AKATSUKI, Imamura et al., 2017).

It was not until the mid-1990s that scientists began to apply RO techniques to the Earth's atmosphere using Global Navigation Satellite System (GNSS) signals as transmitting sources (Kursinski et al., 1996, 1997; Ware et al., 1996). To date,

most Earth GNSS RO observations are taken from low-Earth orbiting (LEO) satellite constellations such as the Constellation Observing System for Meteorology, Ionosphere, and Climate (COSMIC-1, Anthes et al., 2008), the GNSS Receiver for Atmospheric Sounding (GRAS, Luntama et al., 2008) onboard the MetOp satellite series, and COSMIC-2 (Schreiner et al., 2020). More recently, several private companies (e.g., Spire, GeoOptics, PlanetiQ) launched CubeSat constellations that can offer RO soundings with comparable quality to the more sophisticated satellite RO missions (Bowler, 2020). The impact of spaceborne RO profile assimilation on global weather forecasts has been ranked second among satellite measurements (Cardinali and Healy, 2014), with its impact varying depending on assimilation methods (Boullot et al., 2014; Harnisch et al., 2013; Ruston and Healy, 2020).

RO observations are also possible from receivers inside the atmosphere, as opposed to from the LEO receivers in space. One of the more common in-atmosphere RO platforms is an airplane or drone. Airborne radio occultation (ARO) typically uses custom-built receiver payloads onboard a modified aircraft with additional antennae and is capable of significantly higher spatial sampling density than spaceborne RO (Wang et al., 2016) due to their slower velocities compared to the LEO receiver satellites (e.g., COSMIC-1, COSMIC-2). This allows ARO receivers to potentially track more GNSS signals, creating the potential for more frequent, localized occultations (Chan et al., 2021, 2022; Xie et al., 2008). ARO platforms also have the benefit of providing on-demand RO profiles around transient weather events such as mid-latitude or tropical cyclones. ARO is limited primarily by flight restrictions regulated by aviation safety agencies (e.g., U.S. Federal Aviation Administration, European Union Aviation Safety Agency), and by fuel range of the aircraft. In addition, the slower-moving airborne receivers lead to longer occultation duration with increased tangent point drifting distance, potentially leading to larger variations in the sampled atmosphere in comparison with the spaceborne RO.

Early ARO studies developed and tested research instruments as a baseline for in-atmosphere RO, and modified the traditional spaceborne RO retrieval algorithms (Garrison et al., 2007; Healy et al., 2002; Xie et al., 2008; Zuffada et al., 1999). Open-loop signal tracking algorithms were successfully implemented to reduce unwrapping and tracking errors from airborne platforms (Murphy et al., 2015; Wang et al., 2016). Radio-holographic retrieval methods modified for in-atmosphere radio occultations (Adhikari et al., 2016; Wang et al., 2016) have also been implemented to improve upon geometric optics retrievals (Kursinski et al., 1997, 2000) in the moist lower troposphere to reduce multi-path errors.

Another option for an RO observation platform is a high-altitude balloon, but few attempts have been successfully implemented thus far. The Concordiasi Project (Rabier et al., 2010, 2013) is the only field campaign to date during which balloon-borne RO (BRO) observations were targeted as part of the overall research goal. Haase et al. (2012) detailed the proof-of-concept BRO payload and platform design, along with some preliminary results indicating the feasibility of BRO measurements from the Concordiasi field campaign. More recently, Cao et al. (2022) showed that BRO using a custom-built receiver payload aboard a high-altitude Strateole-2 balloon (Haase, J. S. et al., 2018) is also capable of resolving equatorial Kelvin waves. Other remote sensing projects have used balloon platforms for other purposes (e.g., GNSS Reflectometry (GNSS-R), Carreno-Luengo et al., 2016) but GNSS RO payloads on balloon platforms are still otherwise underrepresented.

Balloon-borne RO has many advantages over spaceborne RO and ARO. Like ARO observations, BRO platforms move slowly compared to spaceborne platforms, and are therefore capable of offering high spatial and temporal sounding densities

over targeted regions. Additionally, BRO platforms can remain aloft and collect observations for weeks to months, depending on the design and capabilities of the platforms (Chan et al., 2021, 2022). BRO platforms can also be tactically launched en masse above and around transient weather events such as tropical cyclones and supercell thunderstorms to provide spatially dense sampling of atmospheric thermodynamic profiles inside and surrounding dangerous weather events.

Multiple U.S. federal agencies have also invested in supplementary RO data to support operational weather forecast. The U.S. National Oceanic and Atmospheric Administration (NOAA) is undergoing a multi-year program intended to incentivize commercial participation in data-as-a-service that can be used for improving weather forecasting through data assimilation (National Oceanic and Atmospheric Administration and National Environmental Satellite, Data, and Information Service, 2020). The U.S. National Aeronautics and Space Administration (NASA) also funds a Commercial Smallsat Data Acquisition (CSDA) Program (National Aeronautics and Space Administration, 2022), of which RO profiles are an area of heavy research. Furthermore, low-cost, on-demand RO data would be highly useful for conducting research in the planetary boundary layer (PBL), a targeted observable in the NASA 2017 decadal survey (National Academies of Sciences, 2018). BRO data could be a low-cost alternative or supplement to spaceborne and airborne radio occultation data for PBL sensing.

The remainder of this paper is structured as follows. Section 2 briefly introduces a newly developed, low-cost and scalable commercial-off-the-shelf (COTS) GNSS receiver onboard high-altitude balloon and the associated flight campaigns, along with the detailed description on BRO data and methodology. Section 3 shows the refractivity retrieval process and quality control procedures based on one representative BRO case. Section 4 examines the overall statistics of the BRO observations and provides a preliminary error analysis. Finally, conclusions and future studies are summarized in Section 5.

## 2    Data and Methodology

### 2.1    Balloon-Borne GNSS RO Payload and High-Altitude Balloon Platforms

A detailed description of the balloon-borne GNSS RO payload developed by Night Crew Labs (NCL) can be found in (Chan et al., 2021, 2022) – a summary is presented here. The instrumentation is comprised of two major components: a mission system support component, and a science payload for GNSS RO profiling. The mission system support component is the Balloon Re-Programmable Integrated Computer (BRIC), which is a third-generation flight management computer supporting data logging as well as power management, flight control, and thermal control. The science payload is the GNSS Radio Occultation and Observable Truth (GROOT, which is a first-generation balloon-borne GNSS RO science instrument based on COTS GNSS equipment. The GROOT payload includes a Swift Navigation Piksi Multi GNSS receiver for raw RO measurements (e.g., carrier phase, SNR, and Doppler velocities), a Trimble BX992 dual-antenna GNSS receiver coupled with an inertial navigation system and an L-band GNSS corrections service, in addition to the BRIC flight computer and other ancillary needs. All balloon-borne RO data described in this study were collected from the GROOT payload. The Piksi GNSS RO receiver is particularly noteworthy, as it is about the size of a credit card, which is quite small compared to most other custom-built GNSS RO receivers for in-atmosphere and spaceborne observations.

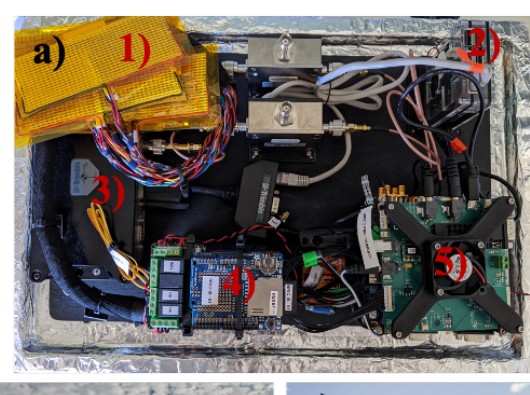

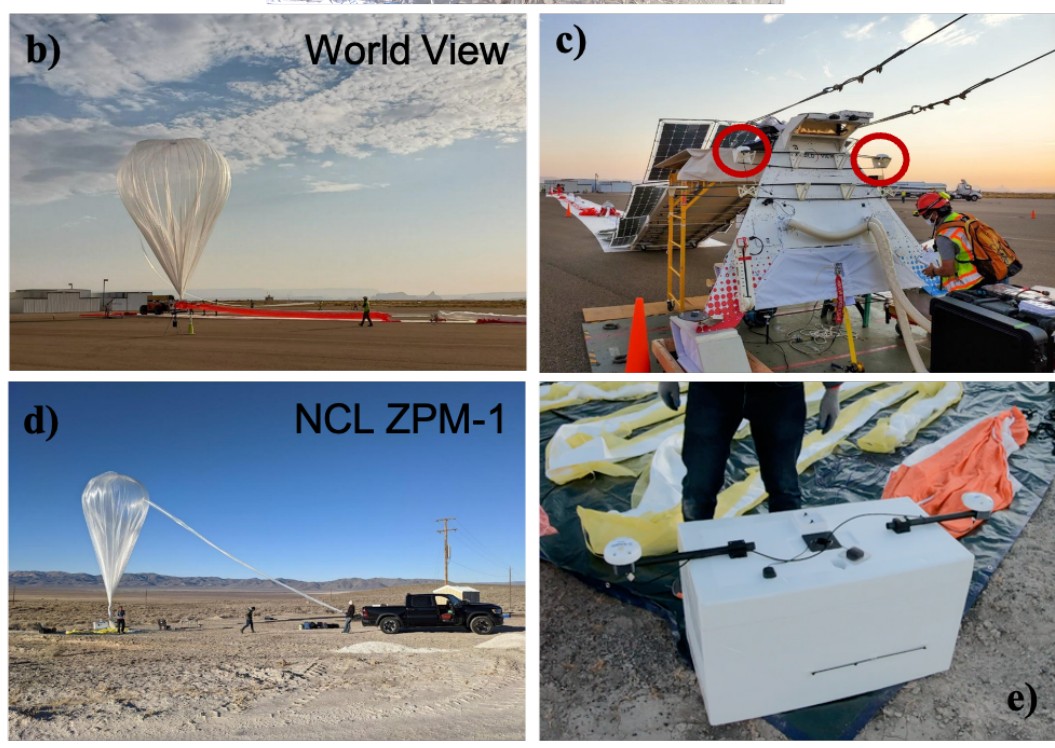

**Figure 1.** a) GROOT payload used in balloon RO flight missions. Numbers indicate the following components – 1) Patch heaters, 2) Raspberry Pi data logger, 3) Trimble BX992 receiver, 4) BRIC flight computer, 5) Power board/Swift Piksi Multi stack for phase and amplitude measurements (adapted from Chan et al. (2022)); b) World View Stratolite balloon being filled prior to launch. c) Balloon RO payload being prepared for launch during World View BRO flight campaigns. GNSS antennae are indicated by red circles. d) NCL ZPM-1 balloon being filled prior to launch. e) ZPM-1 RO payload and GNSS antennae configuration.

In this study, the RO profiles collected from two high-altitude balloon flight campaigns equipped with the GROOT payload were analyzed. Figure 1a shows the GROOT payload as described above, with each component labeled. The first of a series of high-altitude balloon campaigns occurred in August 2020 on a zero-pressure balloon as a secondary payload hosted on the World View Stratolite balloon bus platform (Fig. 1b, c). The World View flight launched out of Page, AZ and maintained 18+ km (60,000+ ft) altitude, enabling five days (120 hours) of continuous data collection. During the flight, GROOT continuously collected balloon state data and RO data from the GPS (United States), GLONASS (Russia), Galileo (European Union), BeiDou (China), and QZSS (Japan) constellations. The World View balloon platform was equipped with yaw control equipment to minimize spinning during flight and to allow for yaw rotation corrections as necessary. After mission termination, the payload was recovered, and the data was processed.

The second flight campaign was the NCL Zero Pressure Balloon Mission 1 (ZPM-1), which launched near Empire, NV on November 28, 2020 (Fig. 1d, e). The ZPM-1 balloon reached a maximum altitude of 31.7 km (104,567 ft), traveling southeast toward Utah. Overnight, ZPM-1 dropped to a lower-than-expected altitude of 17.9 km (59,000 ft) due to colder ambient temperature, which caused the balloon to drift eastwards towards the Rocky Mountains and led to early termination of the mission after 12 hours. During the flight, GROOT collected balloon state data and RO data from GPS, GLONASS, Galileo, BeiDou and QZSS constellations. The payload was later recovered in southern Utah.

## 2.2 Balloon-Borne RO Measurements

Figure 2 shows the ground tracks for the two balloon flight campaigns along with the predicted occultation tangent point locations at the lowest elevation angle, labeled by their respective GNSS RO satellites. The selected occultation events extracted for analysis are highlighted for both flight campaigns. The GROOT receiver can potentially track all currently operational GNSS satellite constellations such as GPS, GLONASS, Galileo, BeiDou, and QZSS. Out of a total number of 680 predicted occultation soundings for the World View flight (based on balloon and GNSS real trajectories), approximately 71% of observations came from non-GPS occultations and were not analyzed (see Fig. A1). This decision was made because GROOT processing on non-GPS satellites data from several previous flights consistently resulted in poor quality ROs as a result of receiver bandwidth limitations. In addition, the closed-loop tracking receiver in the GROOT payload can only track setting (vs. rising) occultations, filtering out another 50% of the available occultations. Of the remaining occultations, only the RO events having measurements with 1) minimum elevation angle less than $0°$, and 2) excess phase greater than 50 meters were processed. This subset of occultations were divided into those that are good quality and those that require additional quality control such as cycle-slip corrections (Fig. A1). Of the original 680 occultations from the World View flight, only 15 cases were extracted for analysis (see Appendix A). Cycle slip corrections were required for 7 of the 15 cases, and one World View case was removed after failing quality control procedures. The same pre- and post-processing algorithms were applied to ZPM-1 measurements. Unfortunately, the ZPM-1 flight encountered power failures during the segments of the flight, resulting in the loss of altitude and a much shorter flight time with fewer occultations being collected, and only 8 occultations out of 84 were selected for processing.

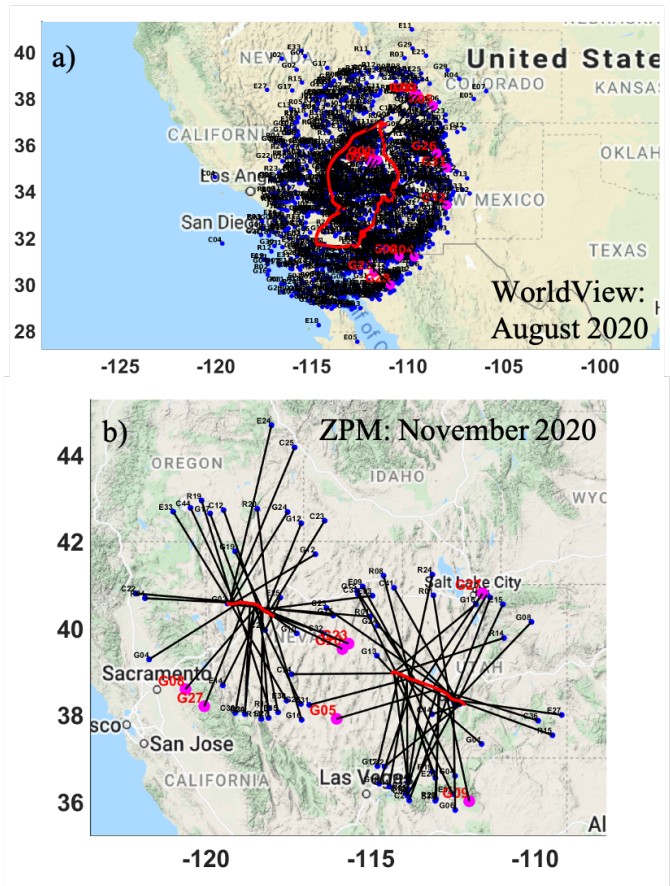

**Figure 2.** High-altitude balloon flight trajectory (red) and predicted occultation tangent point drifting paths and their respective GPS satellites (black) for a) World View and b) ZPM-1. Pink circles with red text indicate the selected RO cases and the occulting GPS satellite number presented in this study.

## 2.3   Balloon-Borne RO Data Processing and Retrieval Methods

After the flight data is logged, several pre-processing steps are required prior to retrieving bending angle and refractivity. The full retrieval process is detailed in (Chan et al., 2022), but is summarized here. The first step in the BRO retrieval is to pre-process the L1 frequency data for ingestion into the retrieval algorithms. Raw line-of-sight (LOS) GNSS observables, along with satellite ephemeris data, and balloon state (position/velocity) data, all of which are measured at 10 Hz, are extracted from payload storage. Once the LOS data is parsed and aligned, the next step is to compute the excess phase by subtracting the

receiver's measured phase from the LOS geometric distance between the occulting GNSS satellite and the receiver. Step three is receiver clock calibration, where the excess phase is calibrated by differencing the excess phase of the occulting and high-elevation GNSS satellites. Step four is cycle slip correction, where an exponential curve is fitted to the data to both smooth and

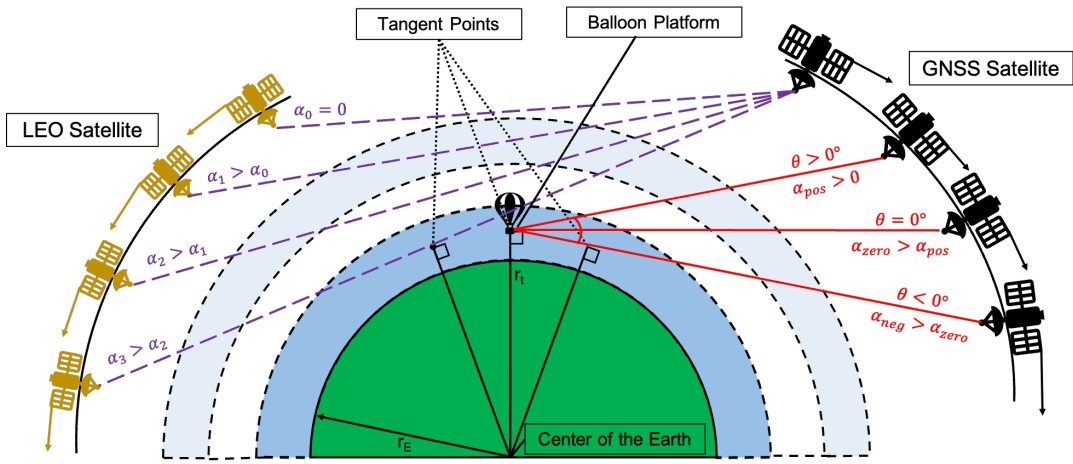

**Figure 3.** Balloon radio occultation transmitter/receiver geometry diagram without showing the bending of the GNSS ray paths. $r_E$ and $r_t$ represent the radius of the earth and the radius at the tangent point, respectively. Adapted from Xie et al. (2008).

remove medium to large discontinuities. Step five uses a Gaussian Process Regression (GPR, Shi and Choi, 2011) to further remove any smaller discontinuities as needed in processing the individual cases. Steps three through five clean and smooth
the excess phase data so that it can be used in the retrieval downstream in the retrieval processing. Finally, the pre-processed results, RO excess phase, excess Doppler shift, SNR, and balloon state, are exported as a NetCDF file, a format convenient for downstream GNSS RO retrieval processing. In our study, the L2 frequency data was often less reliable for retrievals as it would often drop out sooner than the L1 data. However, the L2 data was still used for pre-processing data corrections (e.g., data verification, cycle-slip corrections).
Unlike the well-known traditional spaceborne RO, balloon-borne (and airborne) RO have a more unique, less common transmitter/receiver geometry (Figure 3). The receiver onboard the balloon platform is shown in the center at a radius $r_t$. High-elevation GNSS satellites, where the elevation angle is required to be $\theta > 55°$, are used to calibrate the excess phase at the in-atmosphere receiver. GNSS satellites with elevation angle $-5° < \theta < 5°$ are used in the retrieval process to calculate the bending angle retrievals and subsequent products (Chan et al., 2022; Xie et al., 2018).
To evaluate the performance of the balloon-borne GNSS-RO retrieval algorithm, an end-to-end simulation and retrieval processing system (Figure 4) originally developed for aircraft-based GNSS RO (Xie et al., 2008) was adapted for balloon-borne RO measurements. The processing system includes four main components: (a) a geometric optics ray-tracer, i.e., Radio Occultation Simulations for Atmospheric Profiling (ROSAP, Høeg et al., 1996), which simulates the GNSS RO signal and calculates the associated excess phase, excess Doppler shift, as well as the along-path accumulated bending angle at each
impact parameter as it travels through a prescribed Earth's atmosphere (with either spherical or oblate Earth) given by a 1-dimensional atmospheric refractivity profile. The oblate Earth option is used in our study; (b) a module that derives the bending angle from the LOS excess phase/Doppler measurements through a modified geometric-optics (GO) retrieval, i.e.,

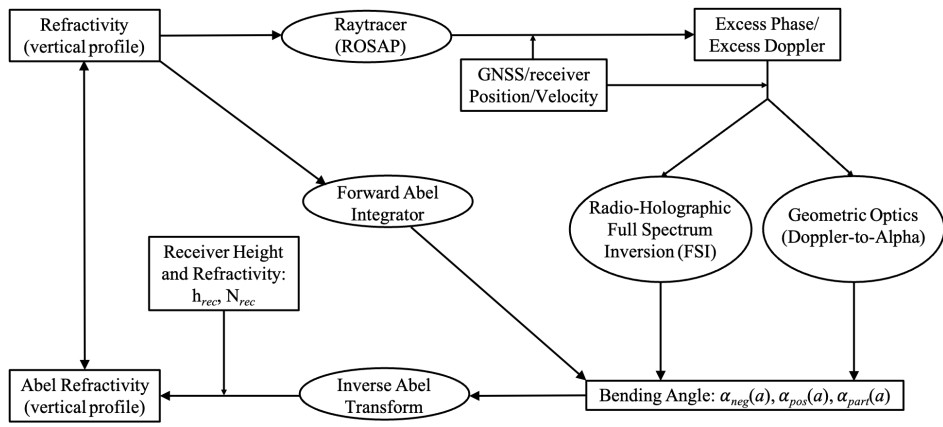

**Figure 4.** Flow chart of the end-to-end simulation and retrieval processing system for airborne and balloon-borne GNSS radio occultation.

Doppler-to-alpha, (Lesne et al., 2002; Vorob'ev and Krasil'nikova, 1994; Xie et al., 2008) and radio-holographic retrieval (i.e., FSI, Jensen et al., 2003) modified for in-atmosphere retrievals (Adhikari et al., 2016); (c) a forward Abel integrator (FAI)

modified for in-atmosphere RO retrievals that generates the bending angle profile through the forward integration of an input refractivity profile (Fjeldbo and Eshleman, 1969; Fjeldbo et al., 1971; Xie et al., 2008); and (d) an inverse integrator that retrieves a refractivity profile via an Inverse Abel Transform (IAT, Fjeldbo and Eshleman, 1969) modified for in-atmosphere RO (Healy et al., 2002; Xie et al., 2008). Note that the GO retrieval method has limited vertical resolution and encounters multipath problems in the moist lower troposphere. Therefore, the FSI method adapted for airborne and balloon-borne RO

retrieval will also be used (Adhikari et al., 2016).

     During the retrieval process, the pre-processed 10 Hz LOS observations are log-linearly smoothed using a 30 second moving window to further ensure that high-frequency noise has been removed. The ROSAP simulation also runs its ray-tracing at the same sampling interval as the provided satellite/receiver geometry. The excess phase output of the ROSAP simulation is also run through the same log-linear smoothing algorithm as the LOS excess phase.

After the bending angle ($\alpha$) profiles are retrieved using either GO or FSI methods, the partial bending angle ($\alpha_{part}$), i.e., the difference between negative and positive elevation bending angles ($\alpha_{neg}$ and $\alpha_{pos}$, respectively) at the same impact parameter, can be derived. The partial bending angle profile is then converted to the refractivity profile using the modified IAT, which requires *a priori* knowledge of the refractivity at the receiver during the occultation event. In addition, the local radius of curvature of the Earth ($r_E$) is also required for conversion of bending angle impact parameter to geometric height as part of

the IAT. Due to the very low excess phase, the raw observation of $\alpha_{pos}$ as well as the $\alpha_{neg}$ near the receiver height is generally very small and can also be rather noisy, which would lead to large errors in partial bending angle and the following refractivity retrieval. Therefore, in this study, the $\alpha_{pos}$ and the $\alpha_{neg}$ within 1.5 km of the receiver altitude were substituted by the simulated bending angle (e.g., FAI or ROSAP bending angle simulation) based on a colocated refractivity profile (Adhikari et al., 2016;

Xie et al., 2018). By doing this, the top 1.5 km of the $\alpha_{pos}$ and the $\alpha_{neg}$ profiles are significantly less noisy, preventing failure
of the IAT.

## 2.4 ERA5 Model Reanalysis Data

To evaluate the quality of balloon-borne RO measurements, we use 3-hourly ERA5 model reanalysis (Hersbach et al., 2020) from the European Centre for Medium-range Weather Forecasting (ECMWF) to provide estimates of atmospheric conditions near the BRO sounding locations. The native ERA5 model grid has $0.25°$ x $0.25°$ horizontal resolution and 137 vertical levels. The ERA5 profiles are referenced to geopotential heights, which are converted to geometric heights for direct comparisons to BRO profiles. The atmospheric refractivity profiles can be derived from the gridded temperature, water vapor pressure (humidity), and pressure profiles according to Equation 1.

$$N = a_1 \frac{P}{T} + a_2 \frac{P}{T^2} \tag{1}$$

Equation 1 is an approximation of atmospheric refractivity for the neutral atmosphere where N is refractivity in [N-units], P is atmospheric pressure in [hPa], T is atmospheric temperature in [K], Pw is the water vapor pressure in [hPa], and $a_1$ and $a_2$ are unitless coefficients with values of 77.6 and $3.73x10^5$, respectively (Kursinski et al., 1997, 2000).

In addition, during an occultation event, the tangent point (TP) is located at the receiver position, when the occulting satellite is at the zero elevation. In-atmosphere RO bending angle retrieval requires *a priori* atmospheric refractivity at the receiver, which can be provided by a colocated ERA5 profile, when high-quality in-situ measurements are not available (Xie et al., 2008, 2018; Adhikari et al., 2016). Thus, for simplicity, each occultation event uses one referencing refractivity profile from the colocated ERA5 grid at the zero-elevation ($\theta = 0$) angle tangent point location. This refractivity profile is used to compute the time series of refractivity at the receiver throughout the occultation observations by interpolating the refractivity profile to the receiver height at each time stamp. Furthermore, considering the high horizontal resolution of ERA5 reanalysis and the potential fine scale variations of refractivity that are smaller than RO horizontal footprint, we use a median refractivity profile of a $1°$ x $1°$ horizontal grid surrounding the zero-elevation TP location for input into the initial ROSAP and FAI simulations. The tangent point locations during the occultation event can therefore be derived from ROSAP ray-tracing simulation with the real occultation geometry. It is important to consider the relatively large horizontal drift of BRO TPs when determining the accuracy of the retrieved refractivity profiles. To best evaluate the quality of the individual retrieved BRO refractivity profiles, the final refractivity comparison uses the ERA5 profile at the 5 km TP location determined by the initial ROSAP simulations. Therefore, three separate refractivity profiles from ERA5 are used during the retrieval process: the zero-elevation angle refractivity profile for the time series of refractivity at the receiver, the $1°$ x $1°$ median refractivity profile surrounding the zero-elevation angle location used for input into simulations, and the 5 km TP location refractivity profile used for final retrieval comparisons.

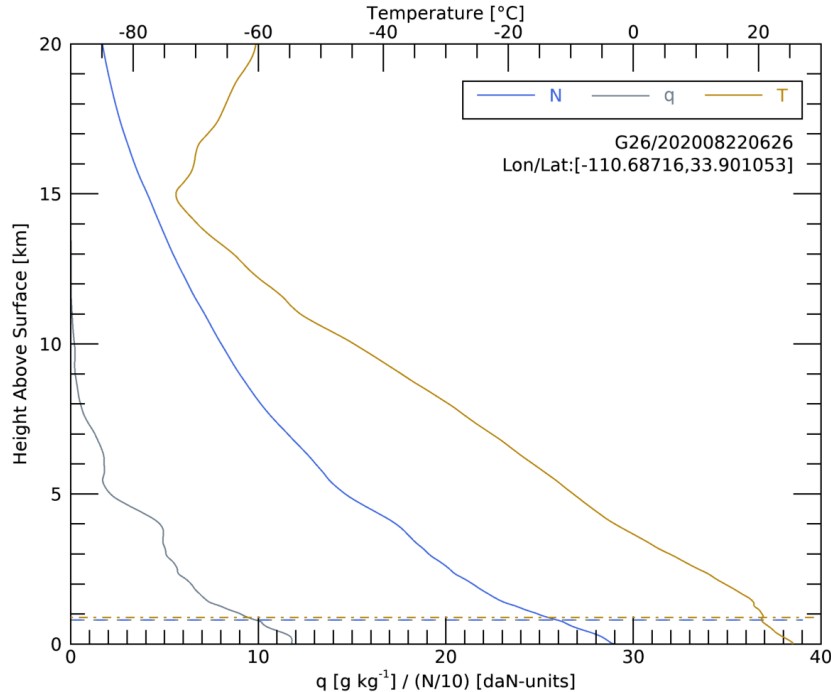

**Figure 5.** ERA5 vertical profiles of temperature (*T*, gold), specific humidity (*q*, grey), and refractivity (*N*, blue) at 0600 UTC colocated with the World View G26 BRO case on August 22, 2020. Note that for display purposes, one-tenth of the refractivity (N/10) was plotted. In addition, the colocation longitude and latitude of the BRO profile are indicated. Dashed lines indicate the planetary boundary layer height (PBLH) as determined by the gradient method from each variable (see text for details).

## 3 Case Study: BRO from the World View Campaign

Here we focus on one typical BRO sounding, an approximately 27-minute-long BRO measurement from the World View
campaign (hereafter referred to as WVG26) at 06:26 UTC, August 22, 2020. The WVG26 case occurred over the Tonto National Forest, northeast of Phoenix, Arizona. The colocated (zero-elevation angle location) ERA5 thermodynamic profiles (temperature, specific humidity, and refractivity) at the location of this BRO sounding with tangent point at 5 km above mean-sea-level are shown in Figure 5. During the occultation event, the local atmosphere was hot and very dry, particularly above 5 km. Additionally, the planetary boundary layer height (PBLH) detected using the gradient method (Ao et al., 2008, 2012;
Nelson et al., 2021; Winning et al., 2017) for this case was found to be at approximately 0.9 km, clearly marked by a distinct temperature inversion (gold, dash-dot line) and weak vertical gradients in specific humidity (grey, dotted line), and refractivity (blue, dashed lines) shown in Fig. 5). The cold-point tropopause was located around 15 km altitude.

Figure 6a shows the WVG26 calibrated excess phase observations (blue) alongside the excess phase from the ROSAP simulation (red) and the difference between the LOS observations and the ROSAP simulation (black dashed line). The calibrated

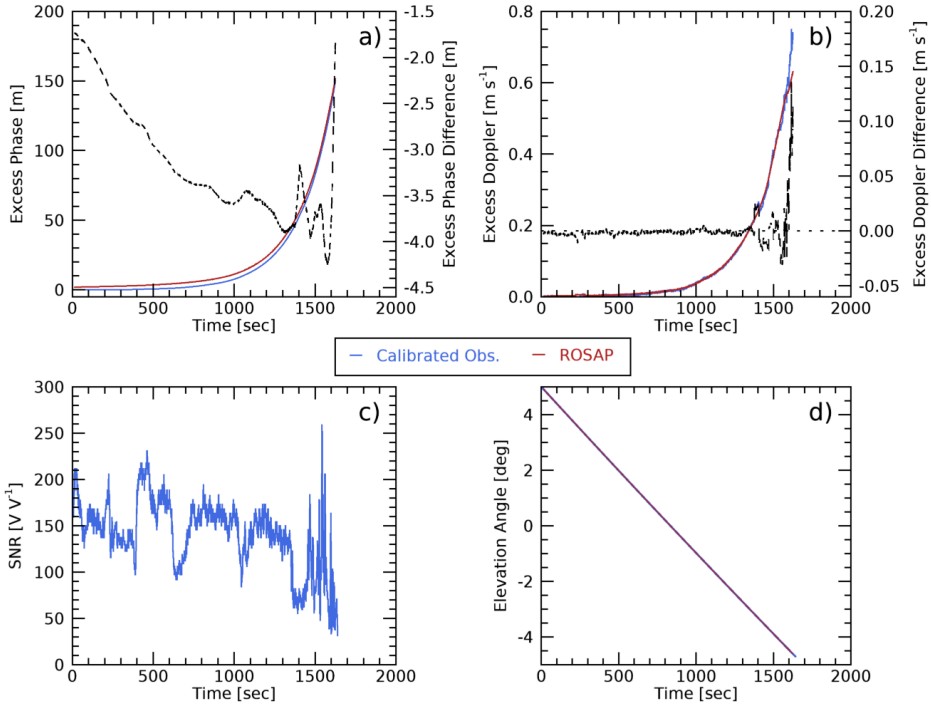

**Figure 6.** WVG26 time series of a) excess phase, b) excess Doppler, c) signal-to-noise ratio, and d) elevation angle from calibrated observations (blue), and ROSAP ray-tracing simulation (red). Dashed lines in panels a) and b) represent differences between the LOS observations and the ROSAP simulation.

excess phase delay compares favorably with the corresponding ROSAP excess phase, with differences between the two within three meters of each other throughout the whole time series. The high consistency throughout the time series is also seen in the excess Doppler comparison (Fig. 6b) with differences only exceeding $0.01 ms^{-1}$ at the end of the time series. The observed L1 signal-to-noise ratio (SNR) for the WVG26 case is shown in Figure 6c. As the signal penetrates deeper into the atmosphere (i.e., the elevation angle dipping below the local horizon to approximately –4.5°, Fig. 6d), the SNR typically decreases and becomes much more variable due to high signal dynamics resulting from the fine moisture variations in the lower troposphere. The overall mean L1 SNR from the GROOT receiver (141.79 $VV^{-1}$) is smaller than the mean SNR from the COSMIC-1 and SAC-C GNSS RO satellite missions (approximately 700 $VV^{-1}$, Ao et al., 2009; Ho et al., 2020). Although the L1 SNR values from the Piksi receiver are approximately 5 times less than the values from spaceborne RO missions, considering the compact size of the Piksi receiver, such high L1 SNR values are quite impressive and can be partially attributed to the smaller defocusing effect compared to that of spaceborne RO as the Piksi receiver is inside the atmosphere.

Figure 7a shows the BRO bending angle profiles from GO and FSI retrievals as a function of impact height (impact parameter minus the local curvature radius of the Earth) for the WVG26 case. Note that as discussed in Section 2.3, the noisy $\alpha_{pos}$ and

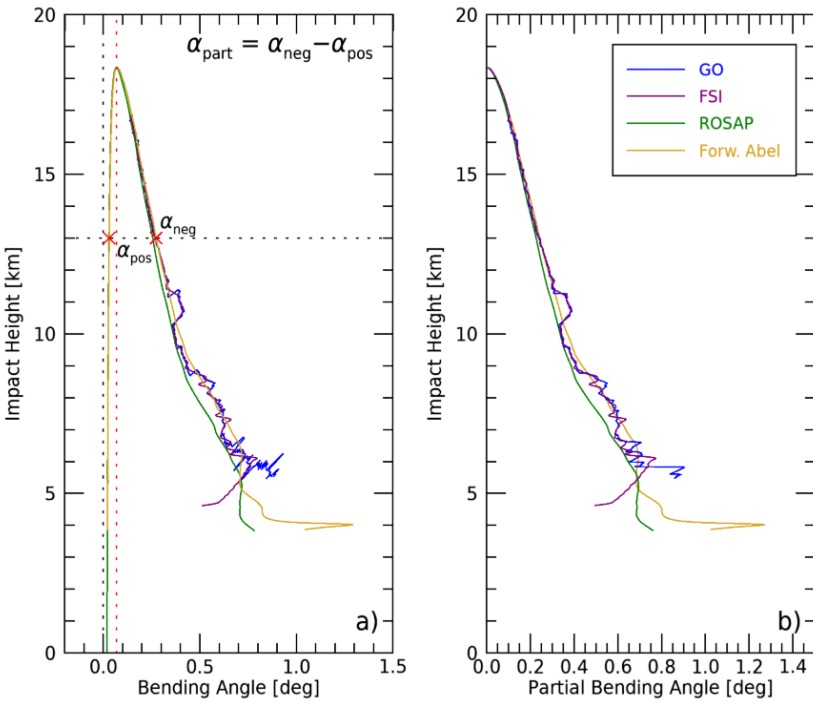

**Figure 7.** a) Bending angle for WVG26 case from the GO retrieval (blue) and FSI retrieval (purple), in comparison with the colocated ERA5 bending angle from ROSAP ray-tracing simulation (green), and forward-Abel-integrator (gold). Conceptual calculation of partial bending angle at 13 km impact height is also shown. b) Final partial bending angle for WVG26 case from the GO retrieval (blue), FSI retrieval (purple), ROSAP ray-tracing simulation (green), and Forward Abel transform (gold).

$\alpha_{neg}$ within 1.5 km below the receiver height were replaced by the simulated FAI bending angle based on the colocated ERA5 refractivity profile (Fig. 5). Fig. 7a also shows a conceptual calculation of the partial bending angle as discussed in Section 2.3. At each impact height, $\alpha_{pos}$ is subtracted from $\alpha_{neg}$ to calculate the partial bending angle ($\alpha_{part}$), which is later used to retrieve the final refractivity profile. Figure 7b shows the partial bending angle calculated using the method shown in Fig. 7a with 100 m log-linear vertical smoothing applied. It is worth noting that BRO bending angle observations from both the GO and FSI retrievals match the colocated ERA5 FAI and ROSAP simulations quite well from the balloon altitude (just over 18 km) all the way down to impact heights of around 6 km (corresponding to approximately 4 km above MSL). Differences between the retrievals and the ROSAP simulation between 10 and 11 km, as well as between 6 and 8 km, are most likely caused by platform yaw instability as discussed in Section 2.1.

Figure 8a shows the refractivity retrieval for the WVG26 BRO case. The 5 km tangent height ERA5 refractivity profiles colocated with the WVG26 case is approximately 425 km from the receiver (zero-elevation) location, and the terrain changes significantly compared to the origin point in the Tonto National Forest in Arizona, USA. From the top of the refractivity profiles,

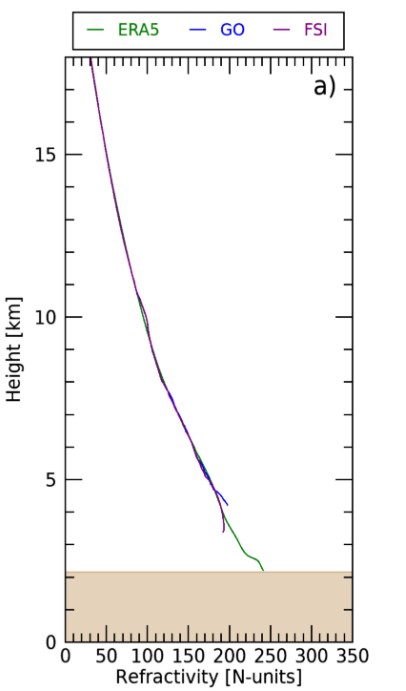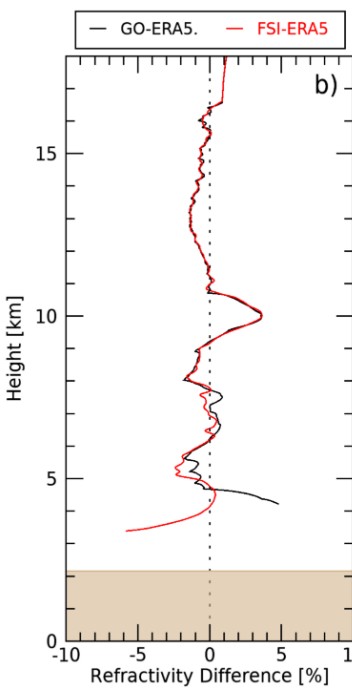

**Figure 8.** a) BRO refractivity retrieval for WVG26 case from GO (blue) and FSI (purple) compared to the colocated refractivity profile from ERA5. b) Fractional refractivity difference between BRO retrieval and ERA5 (GO, black; FSI, red). Local terrain at the location of BRO tangent point at 5 km above MSL is marked by tan polygons.

both the GO and FSI retrievals match the ERA5 refractivity profile from the location of the 5 km tangent height very well. Similar to the $\alpha_{part}$ results, the refractivity retrieval from FSI bending angle reaches about 1 km deeper into the atmosphere than the GO retrieval, highlighting the usefulness of the FSI over GO methods. This is primarily due to the improvement from FSI on resolving the multi-path problem over the GO method in the moist lower troposphere.

In order to quantify the differences between the retrievals and the colocated ERA5 profile, the fractional refractivity dif-
245 ference profile is calculated. Figure 8b shows the fractional refractivity difference between the refractivity retrievals and the colocated ERA5 profile, respectively. The GO retrieval is highly consistent with FSI retrieval above ~7.5 km and start showing small differences below. Both retrievals perform very well in the middle troposphere, between 5 km and 10 km. Once more water vapor is encountered below 5 km altitude, the magnitude of the median difference for the GO retrieval increases. Additionally, intermittent signal degradation and loss due to platform yaw rotation likely induces retrieval errors in the FSI retrieval
and stops the GO retrieval closer to the surface, resulting in sharp changes in *N*-bias for both retrievals.

**Table 1.** Summary statistics for median refractivity differences between GO/FSI retrievals and the colocated ERA5 with corresponding median absolute deviation over varying height ranges from all World View flight campaign BRO profiles.

| Height Range [km] | GO Median $N$ Difference [%] | FSI Median $N$ Difference [%] |
|:---:|:---:|:---:|
| 0-5 | $-5.86 \pm 1.99$ | $-3.60 \pm 3.26$ |
| 5-10 | $-0.25 \pm 2.97$ | $0.32 \pm 3.06$ |
| 10-15 | $0.57 \pm 2.44$ | $1.06 \pm 2.29$ |
| 15-20 | $0.53 \pm 0.71$ | $0.23 \pm 0.94$ |
| Overall | $0.03 \pm 2.28$ | $0.24 \pm 2.61$ |

## 4  Evaluation of Balloon-Borne RO Refractivity Retrievals

Figure 9 shows the fractional refractivity difference between GO/FSI retrievals and the colocated ERA5 profile for each BRO sounding from the World View flight campaign. Individual fractional refractivity difference profiles are calculated by interpolating the retrieval and ERA5 profiles to the same 10 m vertical grid before taking difference between the refractivity profiles.
Summary statistics for all World View flight campaign BRO profiles calculated over different height ranges are shown in Table 1. The median refractivity difference between the GO retrieval and the colocated ERA5 oscillates within 0.25% with MAD between 0.71% above 15 km and approximately 2.28% across all levels (see Table 1). The GO refractivity retrieval starts showing negative $N$-bias below approximately 6 km, with a median refractivity difference of -5.86% (MAD: 1.99%) over the 0-5 km height range. The FSI retrieval also starts showing negative $N$-bias below approximately 6 km, but with a smaller median difference of -3.60% (MAD: 3.26%). The higher magnitude $N$-bias in the lowest portions of the troposphere have a variety of potential causes. The lowest level negative $N$-bias is likely caused by the tracking errors introduced by the closed-loop tracking receiver, which is a well-known problem that could easily degrade the BRO observation quality (Ao et al., 2009; Wang et al., 2013). Additionally, high spatial variations in moisture content can also cause low SNR or high signal dynamics ultimately resulting in a negative bending angle bias (Wang et al., 2016). It is also important to consider that there is likely to be increased low-sampling bias closer to the surface, weakening the robustness of the statistics.

Figure 10 shows the fractional refractivity difference profiles for the ZPM-1 flight campaign from the GO and FSI retrievals. The ZPM-1 refractivity differences are slightly positively biased overall compared to the World View campaign data. The ZPM-1 refractivity differences from both retrieval methods show less variability across all heights in both the median profile and the individual refractivity profiles. The ZPM-1 refractivity differences are also slightly positively biased overall compared to the World View campaign data. The GO retrieval has an overall median refractivity difference of 2.57% (MAD: 1.38%). The maximum $N$-bias from the GO retrieval is 4.05% in the lower troposphere. The FSI retrieval has an overall median refractivity difference of 3.13% (MAD: 2.03%). The World View platform had rotational yaw control capability, whereas the ZPM-1 platform did not – as such, the platform and GNSS antennae were free to spin during high-altitude wind gusts. During RO events, rotational movement likely induced position errors in the dual-antenna navigation system as a result of SNR

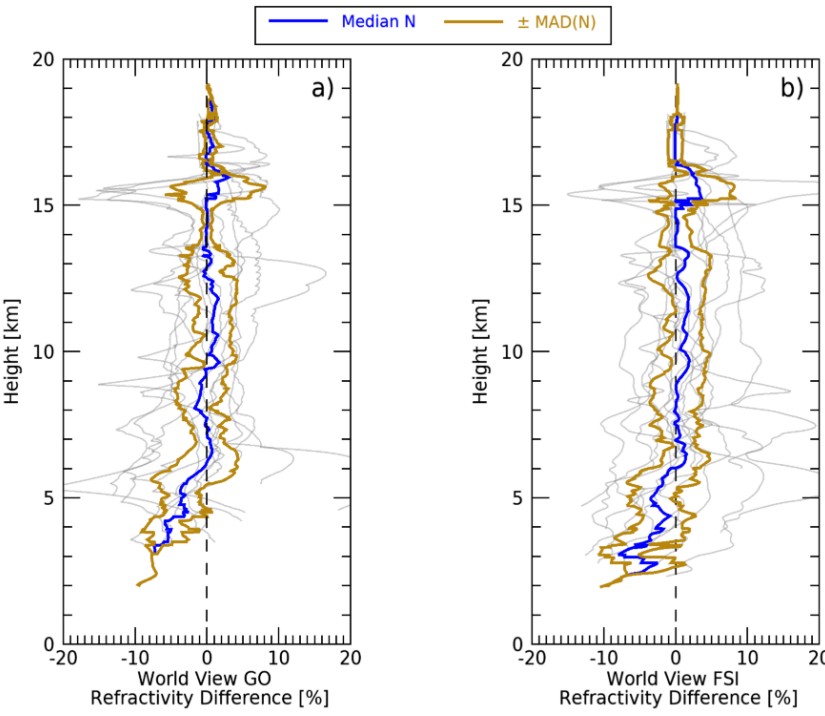

**Figure 9.** Fractional refractivity difference between BRO retrievals and the colocated ERA5 for: a) GO retrievals and b) FSI retrievals of all World View cases (grey). The median fractional refractivity difference profiles are shown in blue and the median ± the median absolute deviation (MAD) is shown in gold

degradation, which could potentially result in larger $N$-bias. Such a phenomenon warrants future investigation of BRO platform control on BRO retrievals.

Errors and bias in BRO refractivity retrievals can come from a variety of potential sources. One potential cause is the difference in precise orbit determination (POD) solutions for BRO missions. Generally speaking, POD solutions for LEO missions aim to have LEO velocity accuracies of $0.5 mm s^{-1}$ or better, and LEO position accuracies of $10 cm$ or better. In contrast, BRO missions are generally capable of velocity accuracies of $30 mm s^{-1}$ or better, and position accuracies of $5 cm$ or better. The larger POD velocity errors are due to difficult-to-model disturbances such as wind gusts and other aerodynamic factors. Xie et al. (2008) showed that the addition of simulated of $5 mm s^{-1}$ random excess Doppler errors will not result in additional $N$-bias, but could possibly introduce less than 1% refractivity error near the receiver ( 10 km) and less than 0.2% below ~6 km. In the case of our study, the larger BRO receiver positioning errors (if random) will likely not introduce significant $N$-bias (Fig. 9) for World View or ZPM-1 cases.

As discussed previously, the limitations of closed-loop tracking receivers may also affect the BRO refractivity retrieval quality. Additionally, low SNR in airborne (in-atmosphere) GNSS RO observations can potentially result in approximately

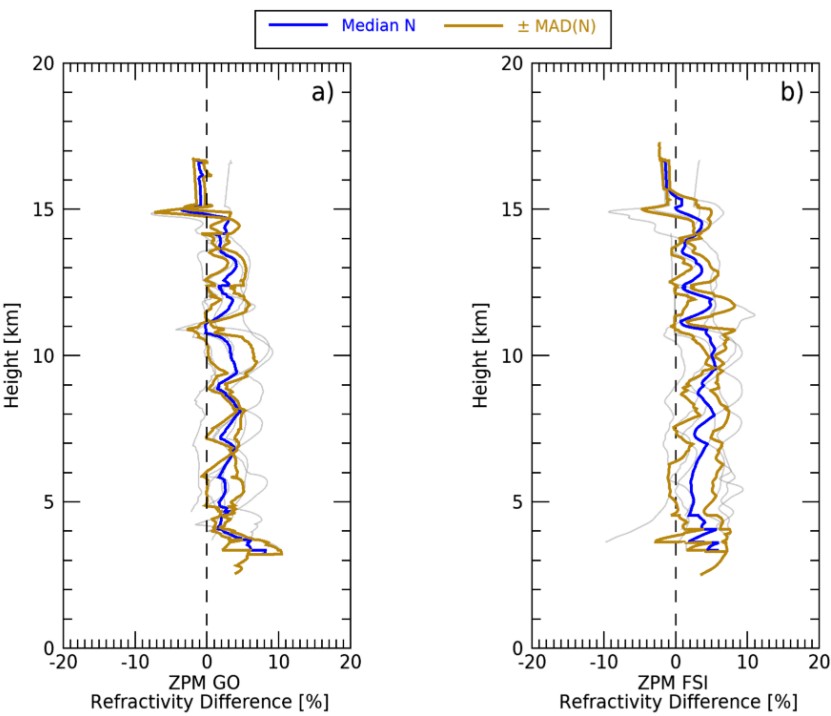

**Figure 10.** Same as Figure 9, but for the ZPM-1 flight campaign.

**Table 2.** Same as Table 1, but for the ZPM-1 flight campaign.

| Height Range [km] | GO Median $N$ Difference [%] | FSI Median $N$ Difference [%] |
|:---:|:---:|:---:|
| 0-5 | $4.05 \pm 0.99$ | $4.01 \pm 2.15$ |
| 5-10 | $2.94 \pm 1.56$ | $3.59 \pm 2.29$ |
| 10-15 | $2.57 \pm 1.78$ | $3.03 \pm 1.73$ |
| 15-20 | $-0.94 \pm 0.64$ | $-1.33 \pm 0.47$ |
| Overall | $2.57 \pm 1.38$ | $3.13 \pm 2.03$ |

$\pm 5\%$ refractivity error (Wang et al., 2016). This estimate is consistent with the overall results showed here, meaning that an improvement to SNR in the lower atmosphere would be extremely beneficial. However, it is important to note that despite
the overall positive bias in the ZPM-1 cases, the median absolute deviation of the cases minimizes in the upper and middle troposphere, much like the World View campaign data. One important caveat for the ZPM-1 campaign data is the small number of available occultations. Furthermore, power loss issues on the platform during the flights caused a decrease in the number of occultations. The low sampling numbers could also lead to larger median refractivity differences.

## 5  Conclusions

In this study, the GNSS RO atmospheric profiling technique has been adapted for use on high-altitude balloon platforms. Most of the past airborne and balloon-borne RO payloads require custom made parts and costly operational expenses that require significant investments. We show the successful implementation of the Night Crew Labs GROOT payload developed from commercial off-the-shelf (COTS) components on high-altitude balloon platforms. This approach is simpler and significantly more affordable than current airborne and space-based methods. The results from the low-cost, highly compact GROOT payload are promising, but more study is needed.

Utilizing a balloon platform for GNSS RO observations has been done only a few times in the past. Haase et al. (2012) showed the proof-of-concept for balloon-borne GNSS RO using a custom-built receiver during the Concordiasi field campaign (Rabier et al., 2010, 2013) and found excess Doppler agreement that would correspond to approximately 1% refractivity difference, indicating that BRO is able to produce high-quality RO profiles. Cao et al. (2022) showed that BRO retrievals can be used to identify equatorial Kelvin waves during long-duration balloon flights with minimal error in the observations. The BRO retrievals from the GROOT payload have refractivity differences in the middle and upper troposphere comparable to previous airborne and balloon-borne RO studies (Adhikari et al., 2016; Haase et al., 2012; Xie et al., 2008; Healy et al., 2002). The added benefit of using BRO platforms is the dense spatial and temporal sampling over targeted regions due to the low platform velocities relative to the LEO-based RO satellites. Additionally, BRO platforms are scalable and can potentially be launched in advance of significant weather events and remain aloft for long periods of time to collect abundant RO observations.

Currently, the major limitation of BRO platforms with COTS payloads is the use of closed-loop tracking GNSS receivers, which limit penetration of RO observations into the lower troposphere due to the large variations in moisture content. Additionally, closed-loop tracking also prohibits the tracking of rising occultations, cutting the potential number of RO soundings in half. For these reasons, design and implementation of COTS payloads capable of open-loop tracking is the next natural step to improving balloon-borne RO. Furthermore, BRO platform orientation can be uncontrolled, so a sudden change in winds aloft can alter the antenna position and cause signal loss. Additionally, the comparatively slow-moving receivers result in longer occultations (approximately 20-30 minutes for one balloon-borne RO event, in comparison to ~1 minute for one spaceborne RO event), which can result in larger unwrapping error (Wang et al., 2016) and lead to further underestimates of bending angle in the moist lower troposphere.

An analysis of the quality of the retrieved refractivity profiles reveals that the overall median refractivity difference for the World View campaign is generally less than 1%. The same analysis of the ZPM-1 campaign data shows that the median is slightly positively biased overall (approximately 3%), but with similar median absolute deviation values. While both GO and FSI retrieval methods offer promising results, it appears that the FSI retrievals tend to outperform the GO retrievals in terms of atmospheric penetration. The limitation of the close-loop tracking could be the primary cause of the negative $N$-bias below 6 km as seen in World View BRO soundings. In addition, the relatively low SNR in the lower troposphere could also lead to negative bending angle bias, which could also be another likely cause of the negative refractivity bias in the lower troposphere.

This study shows that high-altitude balloons with RO payloads can be launched over areas of complex terrain, can potentially remain aloft for far longer than airborne RO platforms, and would hypothetically be deployable in all weather conditions, similar to radiosondes. Furthermore, the balloon-borne RO platform can offer unprecedented high spatial and temporal BRO sampling over targeted regions far higher than traditional spaceborne RO (see Fig. 2). Additionally, balloon-borne RO data could be much more cost effective to retrieve due to the low-cost COTS GNSS RO receiver and overall affordability of the high-altitude balloon flight platform, as the instrument can be retrieved and reused after each deployment. We believe the advances on the COTS GNSS RO receiver development and high-altitude balloon platform control in the future will lead to large increases in high-quality localized BRO soundings over targeted weather events (e.g., severe thunderstorms, tropical cyclones, etc.) and improve regional weather forecasts through data assimilation.

*Data availability.* TAMUCC-derived balloon-borne RO data and retrievals used in this study are publicly available from the NOAA National Center for Environmental Information under CC-BY-NC-SA 4.0 licensing (Nelson et al., 2022). TAMUCC-derived balloon-borne RO data and retrievals are also available upon request by contacting Kevin Nelson at kevin.j.nelson@jpl.nasa.gov. NCL derived balloon-borne RO data and retrievals used in this study are available upon request by contacting Bryan Chan at bryan@nightcrewlabs.com.

All ERA5 data (Hersbach et al., 2020) are available to download from ECMWF and CDS with proper registration and credentials. Instructions for download can be found in: https://confluence.ecmwf.int/display/CKB/How+to+download+ERA5.

## Appendix A: Balloon-Borne RO Cases and Sampling

Balloon-borne RO can collect high density observations around the platform, particularly compared to spaceborne RO. Figure A1 shows a Sankey plot filtering visualization of the World View predicted occultations. Onboard the GROOT payload, duplicate Piksi receivers were used to ensure all GNSS constellations were tracked. Piksi 1 was programmed to log data from GPS, Galileo, Glonass, and Beidou, whereas Piksi 2 was only programmed to log data from GPS and Galileo. As such Piksi 2 was able to log more ROs from GPS. We believe the Piksi receivers and antennae were manufactured such that they were tuned to maximize GPS performance, and as such, satellites from the GPS constellation consistently showed better performance than the other constellations.

Of the original 680 predicted ROs from all GNSS constellations (originally discussed in section 2.1), a total of 485 were incomplete, and therefore not suitable for retrieval processing. The remaining 195 occultations then filter further by removing those from QZSS, Galileo, BeiDou, and GLONASS with low quality due to the frequency tuning inherent to the Piksi receiver. Of the 167 from the GPS system, only 8 good quality cases were ready to use immediately. Another 7 cases required additional pre-processing in the form of cycle-slip corrections. The same process was also applied for cases observed during the ZPM-1 flight campaign.

][h!]

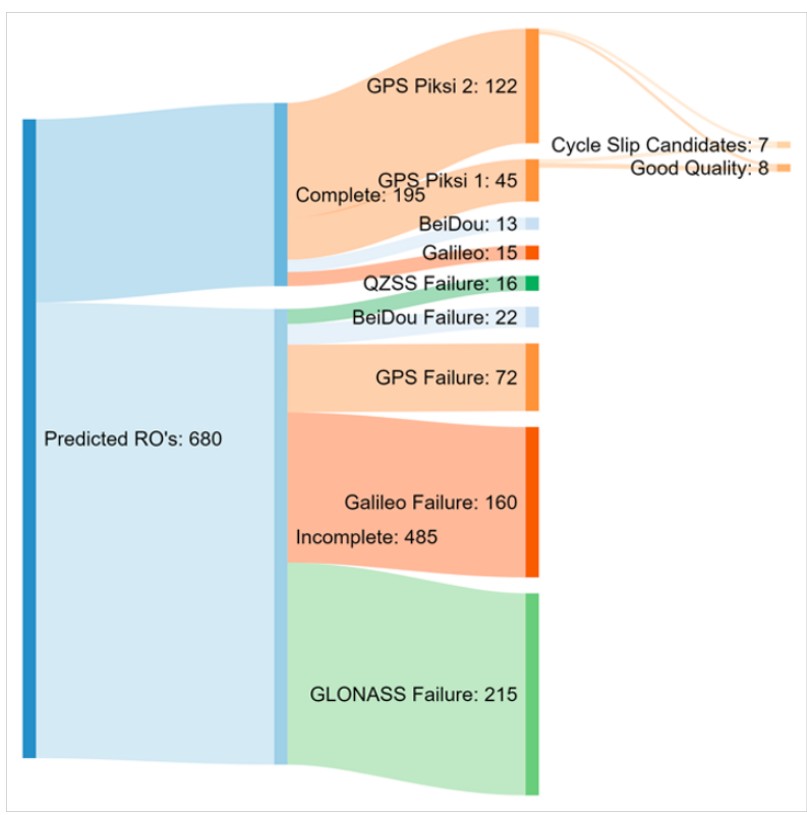

**Figure A1.** Sankey filtering plot visualizing the balloon-borne RO case filtering from the World View campaign.

*Author contributions.* Authors BC and FX developed the concept for the study. Authors BC, AG, TR, PT, JK, and CS completed payload engineering. BC, FX, and AG developed the methodology for the study. KN, FX, JK, and AG developed software used for this study. KN, BC, JK, and AG completed data validation at different steps. Formal analysis was completed by KN, JK, AG, and FX. Data curation was done by KN. Data visualization was done by KN, JK, and AG. Initial drafts of this manuscript were written by KN. Editing and review of the manuscript was completed by KN, FX, BC, and TR. Project administration, funding acquisition, and supervision were done by BC and FX.

*Competing interests.* The authors declare no conflicts of interest or competing interests.

*Acknowledgements.* This paper is supported by NOAA grant 1305M2-19-C-NRMW-0015. Author K. Nelson acknowledges the research assistantship support from Coastal Marine System Science Program at Texas A&M University – Corpus Christi. Authors from Night Crew Labs acknowledge additional support from NASA grant 80NSSC20K0106. This work was done as a private venture and not in the authors'

capacity as an employee of the Jet Propulsion Laboratory, California Institute of Technology. The authors would also like to thank Dr. Loknath Adhikari from the University of Maryland for productive discussion. The anonymous reviewers are also acknowledged for their insight and comments to improve this paper.

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
