# Peer review of "GNSS Radio Occultation Soundings from Commercial Off-the-Shelf Receivers Onboard Balloon Platforms"

_Atmospheric Measurement Techniques, 2022_

## Referee Comment (RC2)

**Manuscript Title**: "GNSS Radio Occultation Soundings from Commercial Off-the-Shelf Receivers Onboard Balloon Platforms"

**Manuscript Number**: amt-2022-198

**Date Reviewed**: 10 OCTOBER 2022

**Manuscript Summary**: This manuscript introduces a novel technique to augment and increase spatial-temporal sampling of existing RO soundings using campaign balloons and commercial off-the-shelf equipment. The balloon ROs are validated against collocated ERA5 reanalysis solutions, and the results show excellent agreement above 7 km with increased biases in the low troposphere.

**Manuscript Evaluation**: The manuscript is well-written, following a logical, and self-explanatory structure presenting clear results and conclusions. Balloon-borne ROs have been assessed, in the past, through NOAA's SBIR program as a possible augmentation to other satellite-borne measurements for increased sampling of the lower atmosphere. **I recommend publication of this manuscript after major revisions, which address all the comments described below.**

**Major Comments – Introduction:**

1. **The introduction lacks science motivation.** Describe the diverse scientific use of the RO data and include statements from e.g., Decadal Survey, NSF, or NOAA documents to demonstrate the need for balloon-ROs.
2. **Schematic of BRO.** Include a simple schematic that shows the location of the tangent point and its drift, as this is not a traditional RO geometry.
3. **Large horizontal drifts.** Mention that ARO/BRO suffer from large horizontal tangent point drifts, resulting in ROs sampling different volumes of air with altitude.
4. **Higher spatial and temporal soundings with ARO and BRO.** Explain how ARO and BRO offer higher sampling than traditional space-borne ROs, if its takes >20 minutes for an ARO/BRO. Do the authors mean higher "*vertical*" sampling?

**Minor Comments – Introduction:**

1. **Line 18: Replace** $CO_2$ with $H_2SO_4$, as S/X-bands are insensitive to $CO_2$.
2. **Line 19: State** the Japanese mission – Akatsuki.
3. **Line 23: Replace** "GRAS" (an instrument) with "MetOp" (a satellite).
4. **Line 25: Remove** "...such as both COSMIC missions."
5. **Line 26: Replace** (Gelaro, 2011) with (Cardinali and Healy, 2014).

**Editorial Comments – Introduction:**

1. **Line 14:** Should read: "Due to vertical atmospheric density gradients, radio...."
2. **Line 16:** Should read: "... before arriving at a receiving antenna on Earth."
3. **Line 44:** Maybe replace "as" with "were"? This sentence does not read well.

**Major Comments – Methodology:**

I feel that the methodology needs to better support the stability and well-constrained errors of the BRO retrievals, in order to demonstrate comparable quality of RO soundings with respect to established missions. For example:

1. **BRIC and GROOT payload details that need clarification:**
   - What is the GROOT sampling rate and does GROOT track pseudo-range?
   - What is the operating payload power?
   - How is the balloon POD solution estimated and what is the accuracy?
   - Why a 50-meter excess phase threshold was selected, and how?
   - What is the vertical altitude range covered during a BRO?
   - Lines 92 & 95: Explain "...*high enough quality*" and "... *poor quality ROs*."
   - Line 113: Do you think you smear out vertical structures by applying GPR?

2. **Impact of Spherical Symmetry Assumption during Balloon-RO:** The spherical symmetry approximation collapses during BRO due to the large horizontal drifts of the tangent point. How do the authors account for it, and what retrieval errors does this assumption introduce to atmospheric products? Are there any other techniques the authors could use to avoid this effect? Or this effect is significantly reduced through differencing of low and high elevation angle during BROs?

3. **Estimation of the Impact Parameter. Line 122:** The impact parameter is estimated simultaneously with the bending angle in Step b). It would be nice, if the authors perform a sensitivity test to quantify bending angle and impact parameter solutions given the balloons POD uncertainty. Unless, the balloons POD solutions are estimated at the same level of accuracy as LEO missions.

4. **Are GO and FSI applied within certain altitude ranges?** E.g., is there an altitude below which the retrieval switches from GO to FSI, or the entire profile is inverted using GO and/or FSI alone?

5. **What is the upper altitude initialization** used in the BRO Abel inversion? It would be nice, if an error propagation analysis, *or relevant discussion*, is included on these topics. **OK, you mention how the refractivity is estimated at the balloon** in Lines 148–159. Maybe it would be better to merge Section 2.4 with 2.3?

**Minor Comments – Methodology:**

1. **Figure 3:** Put arrows at "GNSS/receiver" & "Excess Phase" boxes.
2. **Line 143:** Should read "condition**s**".
3. **Line 154:** Technically, there is no horizontal "resolution". Maybe use "footprint"?

**Major Comments – Results:**

I feel that the results presented in each figure require more discussion to emphasize on the main key points. It appears that BROs are more suited for ROs above 5-6 km altitude (dry regions), due to increased biases with ERA5 down below. Perhaps, the altitude range of BRO performance should also be discussed?

1. The PBL height during WVG26 is at ~0.9 km. **Considering that below 1.5 km from the BRO receiver ERA5 is used to simulate the bending angle** (see Line 138), does this imply that the BRO will never be able to sample PBL height below 1.5 km? This is something that needs to be discussed.

2. **Figure 5a:** It would be nice, if the authors showed the difference in excess phase between the calibrated observations and ROSAP, and then discuss the results with respect to the 50-meter threshold. What altitude does the 50-meter threshold correspond to in WVG26?

3. **Figure 5c:** Explain what causes the SNR jumps of > 100 V/V at different times (e.g., at 335 s, 650 s, etc.)

4. **Figure 6a:** There is ~0.1 deg disagreement between ERA5 and GO/FSI between 6.5 and 8.5 km. Where could this difference come from? Should we trust the BRO or the ERA5? Include some text to describe these differences, and revise the text in Line 185 accordingly.

5. **Figure 7b:** The direction of the refractivity difference changes signs below 4 km. Could the authors explain this?

6. **Figure 8:** The reported refractivity difference exceeds 5% below 5 km. How does this translate to temperature error retrievals with respect to ERA5 temperatures?

7. A separate sub-section discussing various limiting factors on the BRO retrievals (and how to mitigate them) would be a nice addition.

**Minor Comments – Results:**

1. **Line 175:** BRO SNR is 5x smaller than COSMIC and SAC-C. Consider revising.

---

## Author Comment (AC1)

**Author Responses to Reviewer Comments on**: *GNSS Radio Occultation Soundings from Commercial Off-the-Shelf Receivers Onboard Balloon Platforms*

by Kevin J. Nelson[1], Feiqin Xie[1], Bryan C. Chan[2], Ashish Goel[2], Jonathan Kosh[2], Tyler G. R. Reid[2], Corey R. Snyder[2], and Paul M. Tarantino[2]
[1]Texas A&M University - Corpus Christi, Corpus Christi, TX
[2]Night Crew Labs, LLC, Woodside, CA

**We thank all reviewers for their detailed and constructive comments and suggestions. We have made efforts in addressing all reviewers' comments and suggestions and incorporating them into the revised paper. Below is our point-by-point response to each of the reviewers' comments.**

**Comments from Reviewer #1:**

General comments:
The paper shows the retrieved bending angle and refractivity from two balloon-borne radio occultation (RO) campaigns using low-cost commercial off-the-shelf GNSS receivers. Balloon-borne RO could become an important source of information for regional weather observations and forecasts. However, it turns out that there are not many useful occultations in this study (only about 20), and the statistics shown should therefore be taken with a grain of salt (which I think the authors are aware of). It seems to me that the words describing the results make these measurements better than they actually are. This is reflected in some of the specific comments below. Since this is the first paper describing such results using low-cost receivers, and because balloon-borne RO has several benefits over other methods, I think it could be published with minor revisions.

Specific comments:
1. L3: Could "rates" be omitted here? Normally "sampling rate" means the receiver sampling rate during an occultation (e.g., 50 Hz in spaceborne occultations).
   a. **Thank you for pointing this out. This change has been made in the manuscript.**
2. L7: "... show high-quality refractivity profiles in the troposphere with near-zero median difference (~2.3% median-absolute-deviation) from the colocated ECMWF ERA5 reanalysis data.". I would not consider 2.3% median absolute deviation to be high-quality. Looking at the statistics later on, I would not call it high-quality. Spaceborne RO shows less than 1% median absolute deviation above 5 km. Maybe just leave out "high-quality" here (and in the conclusion).
   a. **Median absolute deviation is the equivalent for the median as the standard deviation is to an arithmetic mean. The median refractivity difference profile from the World View campaign shows a near-zero median between approximately 6 km and 19 km (see Figure 8 and Table 1). Given the size of the receiver and the payload being constructed of commercially available parts, we consider this to be high-quality. Schreiner, et al. (2020) showed COSMIC-2 refractivity difference standard deviations on the order of 2-3% in the lower troposphere (< 5 km) and 1-2% between 5 and 10 km. Between 10 and 20 km, the COSMIC-2 refractivity difference standard deviation was shown to be approximately 1%, slightly smaller than our observed variability estimates. However, in the interest of readability, this sentence has been modified in the text to: "*The balloon-borne RO soundings from the World View campaign show high-quality refractivity profiles between 6 and 19 km with near-zero median difference from the colocated ECMWF ERA5 reanalysis data and variability comparable to spaceborne RO missions (~2.3% median-absolute-deviation).*"**

3.  L9: "... are worth further improvement for dense targeted atmospheric soundings to improve regional weather forecasts." This seems a bit cryptic. Do you mean "... need further improvement ..."?
    a.  **We had intended for this to mean that further developments of GNSS RO payloads using commercially available parts should be pursued. This sentence has been changed in the manuscript to:** *"… COTS RO payloads onboard balloon platforms are worth further engineering and study in order to provide capabilities for dense, targeted atmospheric soundings that can improve …"*
4.  L14: "..., Radio signals" should be "..., radio signals".
    a.  **Thank you for catching this error, it has been corrected in the manuscript.**
5.  L20: "late 1990s". I would say mid 1990s. GPS/MET was launched in 1995.
    a.  **This change has been made in the manuscript.**
6.  L23: COSMIC and GRAS needs to be spelled out.
    a.  **The missions have been spelled out in the manuscript.**
7.  L24: "(e.g., SPIRE, GeoOptics, Planet IQ)" I think it is "Spire" and "PlanetiQ".
    a.  **Thank you for catching these errors in the company names. They have been corrected in the manuscript.**
8.  L40: "... (FSI, Adhikari et al., 2016) and phase matching (PM, Wang et al., 2017)". I think it should be written "... (FSI) (Adhikari et al., 2016) and phase matching (PM) (Wang et al., 2017)". Similar comment can be made when referring to COSMIC and GRAS earlier in the same section, and to ROSAP and IAT later on.
    a.  **Thank you for this comment. However, the authors feel that this style minimizes confusion by including extra parenthetical statements. AMT does not provide guidance on this as part of the manuscript formatting. As such, we have elected to keep this as is so we can remain consistent throughout the article.**
9.  L44: "... as part of ...". Do you mean "... was part of ..."?
    a.  **Thank you for catching this. The sentence has been modified to the following: "…** balloon-borne RO (BRO) observations were targeted as part of the overall research goal.**"**
10. L55: "onboard high-altitude balloon". Should it be plural "... balloons"?
    a.  **Thank you for catching this error. It has been corrected in the manuscript.**
11. L81: "... the NCL Zero Pressure Balloon Mission 1 (ZPM-1) was launched ...". I think a "that" is missing in front of "was".
    a.  **Thank you for catching this typo. It has been changed in the manuscript.**
12. L110: Are there any relativistic effects that need correction? I'm not sure, but would like the authors to think about it if they haven't.
    a.  **Thank you for considering this. We did not perform any corrections for relativistic outside of those already incorporated into the PPP processing and basic retrieval process.**
13. L156: "The tangent point locations during the occultation event can therefore be derived from ROSAP ray-tracing". Is that how it is done in general? Are there other ways?
    a.  **Most RO retrievals are capable of doing this as part of the retrieval process. However, our retrieval code for the ARO/BRO retrievals did not have this capability. Because ROSAP ray-tracing provided tangent point data, we elected not to spend time integrating that feature into our retrieval code. It is expected that future, potentially operational, versions of the software beyond a proof-of-concept status will also calculate the tangent point locations.**
14. L157: "To better evaluate the quality of BRO refractivity profile considering the large tangent point drifting distance, the ERA5 profile at the location where tangent point at a height of 5 km above mean-sea-level (MSL) will be treated as the BRO colocated profile for comparison purposes.". Please correct the grammar and syntax in this sentence.

a. **Thank you for catching this editing error. The sentence has been revised to:** *"It is important to consider the potentially large horizontal drift associated with the tangent point of BRO observations. To better evaluate the quality of the individual retrieved BRO refractivity profiles, the ERA5 profile from the tangent point location at 5 km altitude will be treated as the BRO colocated profile for the final refractivity comparison."*

15. L150-159: This part confused me a bit. How many profiles are involved? ERA5 collocated at zero elevation angle tangent point location to compute the time series of refractivity at the receiver by interpolating the refractivity profile to the receiver height at each time stamp throughout the occultation observations. ERA5 median refractivity profile of a 1x1 deg horizontal grid space surrounding the zero-elevation location. ERA5 profile at the location where tangent point at a height of 5 km above mean-sea-level (MSL) will be treated as the BRO colocated profile for comparison purposes. Is that correctly understood? Why not use only one of these for all purposes? Would comparisons become worse if only one was used for all purposes? Please clarify.

   a. **During the retrieval process, two ERA5 refractivity profiles. The first is the median 1°x1° profile used to run the initial ROSAP simulation, run the initial Forward Abel Integrator, and determine the time series of refractivity at the receiver during the occultation. Once the initial ROSAP simulation is complete, we use the tangent point longitude and latitude at 5 km tangent height to determine the final ERA5 profile for comparison of the BRO refractivity retrievals. This has been re-written in the manuscript to be less confusing.**

16. Figure 4: What are the dashed lines slightly below 1 km? There are two of them. Please mention in caption. The y-label: 'Height [km]' is cut off. Is this height above surface (not MSL that far inland, I suppose)? I think it should rather be [10 N-units] (or [da N-units], da means deca) instead of [(1/10) N-units] on the lower axis.

   a. **Thank you for catching this. The y-axis label was cut off because of a LaTeX formatting error, which has been fixed, in addition to clarification that the height is above surface. Your suggestions for the x-axis label have also been incorporated to show dN-units (deci N-units) instead of (1/10) N-units. The dashed lines indicate the boundary layer height detected with the gradient method based on temperature, specific humidity, and refractivity (Nelson et al., 2021; Ao et al., 2012, 2008; Winning et al., 2017; Xie et al., 2006). This was discussed on line 165-167. However, this sentence has been re-worked to call attention of the PBLH lines more effectively.**

17. L168: "The calibrated excess phase delay compares favorably with the colocated ERA5-derived excess phase delay". That is difficult to see. The two curves look quite different to me. Is it necessary to say this?

   a. **Thank you for this comment. We suspect that the reviewer was somewhat confused on their assessment of this panel of Figure 5 and comparing the calibrated excess phase delay (red) with the raw observations (black). Part of the calibration involves receiver clock calibration by subtracting the high-elevation satellite's excess phase delay from the raw occultation GNSS observations. When comparing the calibrated and simulated (green) curves, the differences are minimal.**

18. Figure 5: Here is shown excess phase. Is there a difference between 'excess phase' and 'excess phase delay'? I think they are the same, and I'm not sure what delay means in this context. Maybe "delay" could be omitted in the text.

   a. **Thank you for pointing this out. "Excess phase" is shorthand for "excess phase delay." We have re-written this section to be clearer that they are interchangeable.**

19. L170: "... likely due to satellite-receiver geometry fluctuations." Why would that only affect the ray tracing and not the observations? I would think that the reason is rather the steps taken to clean and smooth the excess phase data, which makes the observed excess Doppler smooth compared to

the ray tracing. I suspect that the variations seen in Fig. 5b come from numerical noise in the ray tracing in combination with taking the derivative to obtain the excess Doppler (it takes only a few mm noise to create excess Doppler noise of the size seen in Figure 5, given a sampling rate of 50 Hz). How would it look if the ray tracing results were smoothed in the same way as the observations? What was the receiver sampling rate?

    a. **Thank you for pointing this out. The receivers for both field campaigns in this study sampled at 10 Hz. While the reviewer is correct that there is likely some contribution to the noise in the excess Doppler from taking the derivative in the excess phase delay and numerical noise from the ray tracing model. ROSAP is highly sensitive to small changes in the transmitter/receiver geometry in such a way that it can induce noise on the scale of millimeters, leading to additional fluctuations in the excess Doppler shift. Running the ROSAP simulated excess phase delay through the same smoothing process as the observed excess phase delay results in the removal of such high-frequency variabilities. Note that Fig. 5b (now Fig. 6b) has been updated to reflect this change.**

20. L174: "(141.79 V/V) is on the same order of magnitude as the mean COSMIC-1 and SAC-C GNSS RO satellite missions (700 V/V) ... this is quite impressive." I think that is a misleading statement. 141 and 700 differ by a factor of 5. That is not the same order of magnitude. Please revise.

    *a.* **Thank you for bringing this to our attention. While the SNR values from the Piksi receiver are, in fact, on the same order of magnitude (order of 100), they are, indeed less than the spaceborne RO SNR values by a factor of 5. This sentence has been re-written to better clarify the difference between them and why the SNR is impressive:** *"The mean SNR from the GROOT receiver (141.79 V V$^{-1}$) is on the same order of magnitude (order of 100) as the mean SNR from the COSMIC-1 and SAC-C GNSS RO satellite missions (approximately 700 V V$^{-1}$, Ao et al., 2009; Ho et al., 2020). While the SNR values from the Piksi receiver are still approximately 5 times less than the values from spaceborne RO missions, the compact size of the Piksi receiver makes such high SNR values quite impressive."*

21. Figure 5: Why is there a time jump in the observations in Figure 5d? Please discuss in the text.

    a. **Thank you for catching this problem. The reason behind the jump in the time in Figure 5d was a plotting error. The error has been addressed and the updated figure is now in the manuscript.**

22. Figure 6: How is the ROSAP bending angle calculated? From simulated excess Doppler? Or ray traced bending angle? Please discuss in the text.

    a. **The Radio Occultation Simulations for Atmospheric Profiling (ROSAP, Høeg et al., 1996) is a 3-dimensional raytracing simulation suite. Given an initial estimate of atmospheric refractivity and transmitter/receiver geometry, ROSAP iterates along a ray path determined by a model and accumulates the bending angle along the occultation ray path. In some cases, ROSAP is unable to converge on a solution due to multipath problems resulting from sharp gradients in the atmosphere, though this has been alleviated with more recent versions of the software. Some clarification has been added to the introduction of the ROSAP simulations in Section 2.3.**

23. L194: "... difference between the GO/FSI retrievals and the colocated ERA5 profiles, respectively." What does "respectively" refer to in this sentence? Are there more than one ERA5 profile? Please revise.

    a. **Thank you for this comment. In this sentence, respectively refers to the refractivity retrievals (GO/FSI) and the colocated ERA5 profile. The sentence now reads:** *"Figure 8b shows the fractional refractivity difference between the refractivity retrievals (GO and FSI) and the colocated ERA5 profile, respectively"* **in order to make the relationship clearer.**

24. L196: "Above 10 km, the median differences between the retrievals and the colocated 5 km tangent height ERA5 profile are both -0.27% to -0.26% with a median-absolute-deviation (MAD) of approximately 0.62%.". I don't see that in Figure 7b. Looks like the median differences are about -1.5% to 4%. Are you taking a mean over some height range? Please clarify.
    a. **Thank you for this comment. Figure 7a (now 8a with a figure added earlier on) shows a single refractivity retrieval from GO, FSI, as well as the colocated ERA5 profile. Figure 7b (8b) shows the refractivity difference profiles for GO-ERA5 and FSI-ERA5. As these are individual profiles, the median "above 10 km" is a single value calculated from all data points above 10 km as well as within other regions. This section has been modified to better describe the statistics over the individual profiles from each retrieval.**
25. L197-201: I don't see the median/MAD and the numbers discussed near the end of section 3. How can you have median and MAD values for only one occultation? Please clarify.
    a. **As in the previous response, we calculate the median refractivity difference and the median absolute deviation over a range of height values.**
26. L207: "... with the minimum bias with a median difference ..." I don't understand that part of the sentence. Can it be written differently?
    a. **Thank you for catching this. This was likely an editing error that was missed. The sentence has been re-written to the following for clarity: "**The GO refractivity retrieval starts showing negative *N*-bias below approximately 6 km, with a median refractivity difference of -5.86% (MAD: 1.99%) over the 0-5 km height range.**"**
27. Table 1: How are these values calculated? Are they average values over the respective 5 km intervals? Are the data interpolated to common equidistant levels before the averaging? Please explain a bit how they are calculated.
    a. **Thank you for this question. The refractivity difference profiles are calculated by interpolating the retrieval and ERA5 refractivity profiles to a 10 m vertical grid and taking the fractional difference between each one. The median and median absolute deviation profiles are then calculated. The statistical values shown in Table 1 are the median and median absolute deviation values calculated over the specific height ranges given in Table 1. The text has been updated to better explain this process.**
28. L214: "The ZPM-1 refractivity difference shows much more variability ...". I don't see that. In fact, the MAD is generally smaller for the ZPM-1 than for the World View flight campaign, so I don't think the statement is correct. However, since the number of occultations in the statistics is so small (also in the World View data), it is perhaps difficult to conclude if there is a true difference in the variability between the two flight campaigns. I agree that there is a bias in the ZPM-1 data relative to the ERA5 data, but could this just be a coincidence due the small number of occultations? In other words: are results statistically significant?
    a. **Thank you for this comment. What we had intended for "variability" to mean in this context was in reference to the "oscillations" present in both the individual refractivity difference profiles and the median refractivity difference profile. The discussion here has been modified to better describe the oscillating features seen in Figure 9 (now 10).**
    b. **We believe that the small number of samples did have an effect on the statistics, potentially biasing the data. However, we feel that the stronger impact in the bias and oscillatory behavior seen in Figure 9 (now 10) is primarily due to observational errors in the excess phase as a result of the lack of rotational yaw control on the ZPM-1 platform as was described in the text. We have, however, modified this section to make both assessments clearer.**
29. L221: "The limitations of closed-loop tracking receivers may also affect the BRO refractivity retrieval quality. Wang et al. (2016) found that the low SNR ...". Something very similar is mentioned two

paragraphs earlier (L210-212), also with a reference to (Wang et al., 2016). Could the discussion be better coordinated to avoid this repetition? It is not clear to me if it is the same thing or two different things (just with different words and emphasis) that are discussed.

    *a.* **Thank you for this comment. We have modified these sentences as follows to distinguish between potential causes for bias:** *"The lowest level negative N-bias is likely caused by the tracking errors (e.g., cycle slips) introduced by the closed-loop tracking receiver, which is a well-known problem that could easily degrade the BRO observation quality (Ao et al., 2009; Wang et al., 2013). Additionally, high spatial variations in moisture content can also cause low SNR or high signal dynamics ultimately resulting in a negative bending angle bias (Wang et al., 2016)."*

30. L234: Who does "their" refer to here?

    a. **Thank you for catching this. This was a missed editing error, and it has been removed.**

31. L260: "Overall, we show that high-altitude balloons with RO payloads can be launched in all weather conditions ...". I don't think it was shown in this paper. Was there severe weather in the campaigns?

    a. **Thank you for catching this. This sentence has been re-worked to be clearer about how BRO platforms** *can* **be launched into hazardous weather conditions similar to radiosondes.**

32. L284: "... leaving the final 11 cases presented in Section 4". But in Section 2.2 (L100), it say[s] 13 cases. Please clarify.

    a. **Thank you for catching this error. A previous version of the paper mis-counted the available cases in multiple places. The contributing cases have been re-counted and updated in the text accordingly.**

33. Figure A1: Is there a difference between GPS Piksi 1 and GPS Piksi 2? I don't think there was a distinction made anywhere in the text. Is it for the two different campaigns?

    a. **There is no real functional difference between the Piksi 1 and Piksi 2. We had duplicate receivers onboard the GROOT payload. Piksi 1 was programmed to log data from GPS, Galileo, Glonass, and Beidou, whereas Piksi 2 was only programmed to log data from GPS and Galileo. As such Piksi 2 was able to log more ROs from GPS. We believe the Piksi receivers and antennae were tuned to maximize GPS performance, and as such satellites from the GPS constellation consistently showed better performance than the other constellations. This is briefly discussed in the text.**

34. L367: https://doi.org/10.1175/2008BAMS2399.I should be https://doi.org/10.1175/2008BAMS2399.1 (1 instead of I at the end)

    a. **Thank you for catching this error. It has been updated in the text as well as in our citation database.**

**Comments from Reviewer #2:**

Manuscript Title: *GNSS Radio Occultation Soundings from Commercial Off-the-Shelf Receivers Onboard Balloon Platforms*

Manuscript Number: amt-2022-198

Date Reviewed: 10 OCTOBER 2022

Manuscript Summary: This manuscript introduces a novel technique to augment and increase spatial-temporal sampling of existing RO soundings using campaign balloons and commercial off-the-shelf equipment. The balloon ROs are validated against collocated ERA5 reanalysis solutions, and the results show excellent agreement above 7 km with increased biases in the low troposphere.

Manuscript Evaluation: The manuscript is well-written, following a logical, and self-explanatory structure presenting clear results and conclusions. Balloon-borne ROs have been assessed, in the past, through NOAA's SBIR program as a possible augmentation to other satellite-borne measurements for increased sampling of the lower atmosphere. I recommend publication of this manuscript after major revisions, which address all the comments described below.

Major Comments – Introduction:
1. The introduction lacks science motivation. Describe the diverse scientific use of the RO data and include statements from e.g., Decadal Survey, NSF, or NOAA documents to demonstrate the need for balloon-ROs.
    a. **Thank you for this comment. A paragraph has been added in the introduction that details how BRO might be useful for U.S. agency interests.**
2. Schematic of BRO. Include a simple schematic that shows the location of the tangent point and its drift, as this is not a traditional RO geometry.
    a. **Thank you for this suggestion. A geometry figure (Figure 3) has been created and incorporated into the methodology section and is discussed accordingly.**
3. Large horizontal drifts. Mention that ARO/BRO suffer from large horizontal tangent point drifts, resulting in ROs sampling different volumes of air with altitude.
    a. **Thank you for bringing this to our attention. This caveat has been incorporated into the introduction.**
4. Higher spatial and temporal soundings with ARO and BRO. Explain how ARO and BRO offer higher sampling than traditional space-borne ROs, if its takes >20 minutes for an ARO/BRO. Do the authors mean higher "vertical" sampling?'
    a. **Thank you for bringing this up. Additional clarification has been added to the manuscript introduction regarding this. Essentially, the slower airborne receiver velocities relative to LEO satellite velocities allow for the potential for more satellite/receiver pairings that create more frequent, localized radio occultations in a region of interest.**

Minor Comments – Introduction:
1. Line 18: Replace CO2 with H2SO4, as S/X-bands are insensitive to CO2.
    a. **Thank you for catching this error. This correction has been made in the manuscript.**
2. Line 19: State the Japanese mission – Akatsuki.
    a. **Thank you for catching this omission on our part. This has been updated in the manuscript. Satellites have also been added for other planetary RO missions that were missing.**
3. Line 23: Replace "GRAS" (an instrument) with "MetOp" (a satellite).
    a. **This has been clarified in the manuscript: *"To date, most Earth GNSS RO observations are taken from low-Earth orbiting (LEO) satellite constellations such as the Constellation Observing System for Meteorology, Ionosphere, and Climate (COSMIC-1, Anthes et al., 2008), the Global Navigation Satellite System (GNSS) Receiver for Atmospheric Sounding (GRAS, Luntama et al., 2008) onboard the MetOp satellite series, and COSMIC-2 (Schreiner et al., 2020)."***
4. Line 25: Remove "...such as both COSMIC missions."
    a. **Thank you for this suggestion. The manuscript has been updated accordingly.**
5. Line 26: Replace (Gelaro, 2011) with (Cardinali and Healy, 2014).
    a. **Thank you for this suggestion. The references in this line have been updated.**

Editorial Comments – Introduction:

1. Line 14: Should read: "Due to vertical atmospheric density gradients, radio...."
   a. **Thank you for this suggestion. The sentence in question has been updated accordingly.**
2. Line 16: Should read: "... before arriving at a receiving antenna on Earth."
   a. **Thank you for this suggestion. The sentence in question has been updated accordingly**
3. Line 44: Maybe replace "as" with "were"? This sentence does not read well.
   a. **Thank you for this suggestion. This sentence has been updated to the following: "*The Concordiasi Project (Rabier et al., 2013, 2010) is the only field campaign to date during which balloon-borne RO (BRO) observations were targeted as part of the overall research goal.*"**

Major Comments – Methodology: I feel that the methodology needs to better support the stability and well-constrained errors of the BRO retrievals, in order to demonstrate comparable quality of RO soundings with respect to established missions. For example:

1. BRIC and GROOT payload details that need clarification:
   a. **Thank you for bringing this to our attention. Please see our individual responses below.**
   b. What is the GROOT sampling rate and does GROOT track pseudo-range?
      i. **The GROOT receiver records phase/amplitude data and positioning data from separate receivers. We configured GROOT to collect data at 10 Hz. GROOT also tracks pseudo-range.**
   c. What is the operating payload power?
      i. **The operating payload power for GROOT as shown is 20 W. It powers 3 receivers and a high-end computer. If optimized for power in sub sequent payloads, the operating power would be substantially less.**
   d. How is the balloon POD solution estimated and what is the accuracy?
      i. **Balloon POD solution is estimated using the Trimble BX-992 navigation system. It generates position and velocity solutions using multi-band GNSS, MEMS inertial measurement unit, and real-time GNSS corrections via PPP. The accuracy is with 5 cm 3D RMS.**
   e. Why a 50-meter excess phase threshold was selected, and how?
      i. **The 50 m excess phase threshold was chosen to avoid needlessly processing RO profiles that would not penetrate far enough into the middle troposphere. The 50 m threshold was determined partially from the authors' previous experience with in-atmosphere airborne RO and from test flights done with the GROOT payload. According to the specific ARO case in Fig. 6 (a & d), excess phase delay of 50 m at time of ~1400 sec when elevation angle reaches about -3 deg, corresponding to the mid-troposphere or lower in drier atmosphere.**
   f. What is the vertical altitude range covered during a BRO?
      i. **Generally, the vertical altitude range covered during a BRO is small. For example, the WVG26 case detailed in the text only moved between 18.10km and 18.16 km. The balloons used are zero-pressure balloons, so the altitude remains relatively stable. This can, however, potentially not be the case if an occultation is recorded during the initial ascent/final descent, or potentially during inclement weather.**
   g. Lines 92 & 95: Explain "...high enough quality" and "... poor quality ROs."
      i. **In this instance, high enough quality ROs referred to those time series SNR was high enough to extract any data. This section has been re-worked as part of Reviewer #1's comments to better reflect the quality of the data and how the receiver bandwidth affects the potential ROs observed.**

h. Line 113: Do you think you smear out vertical structures by applying GPR?

    i. **As was explained in the text, the GPR is only applied to remove smaller discontinuities in the excess phase time series data, which could significantly improve the data quality. The way the GPR was tuned/applied to the data allowed it to ignore continuous structures, decreasing the likelihood of it affecting any phase variations due to vertical structures. The GPR will smooth out some fine structures in the excess phase. However, we would not expect it to significantly modify the excess Doppler (i.e., derivative of the excess phase), which could directly affect the bending angle retrieval vertical resolution.**

2. Impact of Spherical Symmetry Assumption during Balloon-RO: The spherical symmetry approximation collapses during BRO due to the large horizontal drifts of the tangent point. How do the authors account for it, and what retrieval errors does this assumption introduce to atmospheric products? Are there any other techniques the authors could use to avoid this effect? Or this effect is significantly reduced through differencing of low and high elevation angle during BROs?

    a. **Thank you for asking this question. Firstly, it is necessary to clarify that the tangent point (TP) drifting problem is different from the spherical symmetry assumption limitation. The tangent point drifting does not lead to the collapse of local spherical symmetry (LSS) assumption, as the LSS assumption is applied for each epoch during the occultation event. Both the spaceborne and airborne/balloon-borne RO has the TP drifting problem, with the ARO experiencing larger drifting due to the longer occultation event. However, TP drifting does introduce "representativeness" issues, especially when comparing with in-situ observations (e.g., radiosondes). Therefore, in this study, we use the collocated ERA5 profile at the TP location at 5 km above MSL to get better comparison. The problem can be easily relieved in uses of ARO/BRO in data assimilation.**

    b. **On the other hand, the LSS assumption in ARO events does become problematic, which is the reason for using the positive elevation observation to replace the "missing" half of the RO measurement from ARO receiver to the LEO receiver in spaceborne occultation event (Healy et al., 2002). Note that the calibration of ARO excess phase with high-elevation GNSS satellite is primarily focusing on removing the receiver clock errors, and improve the observation, but do not address the LSS issue. However, with the airborne receiver in the upper troposphere or stratosphere, the majority of the bending contribution comes from negative elevation observations (seen in Fig. 6a, now 7a). For BRO observations, the COTS receiver does not reliably measure the excess phase well for positive elevation due to very small bending introduced by the thin atmosphere above the balloon platform (~18 km). Therefore, we use the simulated positive elevation bending angle given the collocated ERA5 profile (or the known refractivity at the receiver along with the simple exponential refractivity model) to replace the noisy observation (see Section 2c). Such replacement of positive elevation bending could introduce errors but given the very small bending contribution above the receive height (~18 km), we do not expect to see significant impact to the ARO retrieval.**

3. Estimation of the Impact Parameter. Line 122: The impact parameter is estimated simultaneously with the bending angle in Step b). It would be nice, if the authors perform a sensitivity test to quantify bending angle and impact parameter solutions given the balloons POD uncertainty. Unless, the balloons POD solutions are estimated at the same level of accuracy as LEO missions.

    a. **Thank you for this suggestion. In general, POD solutions in LEO missions aim to have LEO velocity accuracies of 0.5 mm/s or better, and LEO position accuracies of 10 cm or better. Whereas the balloon POD solutions have velocity accuracies of 30 mm/s or better, and position accuracies of 5 cm or better. The larger POD velocity errors are due to difficult-to-**

model disturbances such as wind gusts and other aerodynamic factors. Xie et al. (2008) showed that the addition of simulated of 5 mm/sec random excess Doppler errors will not result in additional *N*-bias, but could possibly introduce less than 1% refractivity error near the ARO receiver (~10 km) and less than 0.2% below ~ 6 km. We expect the larger ARO/BRO receiver positioning errors (if random) will not introduce significant *N*-bias (seen in Fig. 8, now 9 for World View cases). However, we do suspect the larger *N*-bias seen in ZPM-1 cases is likely due to positioning errors or biases due to platform yaw control issues. We think the sensitive test could be valuable but would not expect to result in significant improvement of understanding of the retrieval errors. Instead, a brief discussion was added to reflect the concerns of the POD uncertainty.

4. Are GO and FSI applied within certain altitude ranges? E.g., is there an altitude below which the retrieval switches from GO to FSI, or the entire profile is inverted using GO and/or FSI alone?
   a. **Thank you for this comment. No, unlike the application of the transition from GO to FSI near upper troposphere in space RO retrieval to reduce the calculation cost, in this study, both GO and FSI retrieval have been applied throughout the full occultation event. Of course, there is minimum difference between the two retrievals in the upper troposphere and stratosphere, but larger differences exist in the lower troposphere.**

5. What is the upper altitude initialization used in the BRO Abel inversion? It would be nice, if an error propagation analysis, or relevant discussion, is included on these topics. OK, you mention how the refractivity is estimated at the balloon in Lines 148–159. Maybe it would be better to merge Section 2.4 with 2.3?
   a. **Thank you for this comment. The receiver is located at the inflection point in the bending angle profile where the impact parameter maximizes. The in-atmosphere Abel inversion requires *a priori* knowledge of the refractivity at the receiver in order to obtain a correct refractivity profile. This discussion has been slightly modified to remove the accidental discussion of the reanalysis in Section 2.3, leaving the explanation of any use of the ERA5 reanalysis for Section 2.4. For this reason, the authors have elected to leave the two sections separate.**

Minor Comments – Methodology:
1. Figure 3: Put arrows at "GNSS/receiver" & "Excess Phase" boxes.
   a. **Thank you for catching this. The change has been made to the figure.**
2. Line 143: Should read "conditions".
   a. **Thank you for catching this error. It has been**
3. Line 154: Technically, there is no horizontal "resolution". Maybe use "footprint"?
   a. **Thank you for this suggestion. This change has been made to the manuscript.**

Major Comments – Results: I feel that the results presented in each figure require more discussion to emphasize on the main key points. It appears that BROs are more suited for ROs above 5-6 km altitude (dry regions), due to increased biases with ERA5 down below. Perhaps, the altitude range of BRO performance should also be discussed?
1. The PBL height during WVG26 is at ~0.9 km. Considering that below 1.5 km from the BRO receiver ERA5 is used to simulate the bending angle (see Line 138), does this imply that the BRO will never be able to sample PBL height below 1.5 km? This is something that needs to be discussed.
   a. **Thank you for bringing this to our attention. It is not the bottom of the bending angle profiles, but the top, closest to the receiver where the elevation angle is near 0°, where the bending angle is small and very sensitive to RO geometry errors. Doing this significantly reduces the errors from the bending angle retrievals and improves bending**

**angle retrievals in the lower troposphere. We have clarified this section to better explain what portions of the bending angle profiles (both $\alpha_{pos}$ and the $\alpha_{neg}$) are replaced with simulated bending angles.**

2. Figure 5a: It would be nice, if the authors showed the difference in excess phase between the calibrated observations and ROSAP, and then discuss the results with respect to the 50-meter threshold. What altitude does the 50-meter threshold correspond to in WVG26?

    a. **Thank you for this comment. We only use the 50 m threshold based on the past experience to determine whether or not to accept an individual occultation as a "good case" to perform the retrieval process on. According to the specific ARO case in Fig. 5 (now 6) (a & d), excess phase delay of 50 m at time of ~1400 sec when elevation angle reaches -3 deg, corresponding to the mid-troposphere or lower in drier atmosphere. Additionally, Fig. 5a & b (now 6a&b) have been modified to show the differences between the observed and simulated excess phase (a) and excess Doppler and they are discussed in the text.**

3. Figure 5c: Explain what causes the SNR jumps of > 100 V/V at different times (e.g., at 335 s, 650 s, etc.)

    a. **Thank you for this comment. We suspect that the SNR is fluctuating as much as it is due to rotation (yaw) of the World View platform. We think that when the platform rotates out of alignment, it takes time for the yaw-control to kick in to get the platform and receiver back into position. This explains the sharp drops and gradual gains in SNR over time. Smaller jumps are likely caused by higher-frequency/less-extreme platform rotation, in addition to noise or other signal interference. Yaw-control was not included on the ZPM-1 platform, potentially resulting in larger SNR fluctuations.**

4. Figure 6a: There is ~0.1 deg disagreement between ERA5 and GO/FSI between 6.5 and 8.5 km. Where could this difference come from? Should we trust the BRO or the ERA5? Include some text to describe these differences, and revise the text in Line 185 accordingly.

    a. **Thank you for this comment. Simply based on one case, it is hard to draw any solid conclusions about which data to trust as the most accurate representation of the bending angle. The most challenging issue here is that there is no real "truth" because the ERA5 results are from the FAI simulation on model reanalysis data with varying degrees of in-situ assimilation and the BRO is obviously remote sensing. Other than the possible ARO observational error, a likely error source is how representative the ERA5 profile used for the comparison. We believe that Fig. 8 (now 9) might indicate a more solid conclusion that the COTS payload could potentially observe the mid-troposphere with minimal bias, although the number of profiles is still very limited.**

5. Figure 7b: The direction of the refractivity difference changes signs below 4 km. Could the authors explain this?

    a. **Thank you for this comment. The most likely reason for the change in the sign of the refractivity difference below 4 km is observation error in excess phase (no longer accumulating at lower TP heights) induced by closed-loop receiver losing track of RO signal as a result of low SNR likely caused by the platform rotating. As discussed previously in the responses, yaw control measures only activate after a large enough deviation is detected due to wind gusts. The negative biased excess phase leads to negative biased bending/refractivity retrieval.**

6. Figure 8: The reported refractivity difference exceeds 5% below 5 km. How does this translate to temperature error retrievals with respect to ERA5 temperatures?

    a. **Thank you for this comment. In general, 1% fractional refractivity difference corresponds to a corresponding 2 K temperature difference, so 5% refractivity error corresponding to ~10K error in temperature and is of limited scientific value. We kept the bad data with**

**both GO and radio-holographic (FSI) retrieval to demonstrate the large retrieval errors are not likely due to multipath problems (frequently seen in GO retrieval), but instead likely result from bad excess phase observations due to the closed-loop receiver tracking errors. Therefore, it indicates the necessity to use open-loop tracking ARO/BRO receivers for moist lower troposphere sensing in the future.**

7. A separate sub-section discussing various limiting factors on the BRO retrievals (and how to mitigate them) would be a nice addition.

   a. **Thank you for this comment. While we appreciate the reviewer's insights here on adding a separate section, the limiting factors on BRO have been discussed already, and are mentioned again within the discussion section. For these reasons, the authors have collectively elected to keep the sections as they are.**

Minor Comments – Results:

1. Line 175: BRO SNR is 5x smaller than COSMIC and SAC-C. Consider revising.

   a. **Thank you for this comment. This was also brought up by reviewer #1. The following is also seen in the response to their comment: "While the SNR values from the Piksi receiver are, in fact, on the same order of magnitude (order of 100), they are, indeed less than the spaceborne RO SNR values by a factor of 5. This sentence has been re-written to better clarify the difference between them and why the SNR is impressive."**

**Author References:**

[revised manuscript text omitted]

---

## Referee Report (RR1)

**Manuscript Title**: "GNSS Radio Occultation Soundings from Commercial Off-the-Shelf Receivers Onboard Balloon Platforms"

**Manuscript Number**: amt-2022-198

**Date Reviewed**: 19 DECEMBER 2022

**Manuscript Evaluation**: This work could potentially augment space-born ROs by better sampling the lower-to-middle troposphere. The authors have done an excellent job in addressing all my comments from the previous review cycle. I am not satisfied with the status of the current manuscript, and **I recommend publication in its current form**.

However, I would like the authors (if they keep working on balloon ROs) to work on techniques to remove the SNR drops during BROs, as well as perform additional sensitivity studies to understand the influence of velocity accuracy on derived products.

---

## Author Response (AR2)

**Author Responses to Reviewer Comments on**: *GNSS Radio Occultation Soundings from Commercial Off-the-Shelf Receivers Onboard Balloon Platforms*

by Kevin J. Nelson[1], Feiqin Xie[1], Bryan C. Chan[2], Ashish Goel[2], Jonathan Kosh[2], Tyler G. R. Reid[2], Corey R. Snyder[2], and Paul M. Tarantino[2]
[1]Texas A&M University - Corpus Christi, Corpus Christi, TX
[2]Night Crew Labs, LLC, Woodside, CA

**We thank all reviewers for their detailed and constructive comments and suggestions during the second round of review. We have made efforts in addressing all reviewers' additional comments and suggestions and have done our best to incorporate them into the revised version of the manuscript. Below is our point-by-point response to each of the reviewers' comments.**

**Comments from Reviewer #1:**

General comments:

My comments below relate to either new text in the revision or to replies to comments in the first review. There are several places where there are differences between the tracked changes provided by the authors and the revised manuscript. Also, a new figure (Figure 3) was added together with a new paragraph discussing it, but the new text was not marked as new in the tracked changes. I ended up not relying on the tracked changes. Comments below refer to what is in the revised manuscript.

Specific comments:

1. L6-8: I urge the authors to remove the claim in the abstract that the data are 'high-quality' and that the variability is comparable to spaceborne RO missions. In their answer (2a) to my comment on this in the first review, the authors refer to the paper by Schreiner et al. (2020) and write: "Between 10 and 20 km, the COSMIC-2 refractivity difference standard deviation was shown to be approximately 1%, slightly smaller than our observed variability estimates." However, 1% is not 'slightly smaller' than 2.3%. The authors also write in their answer: "Given the size of the receiver and the payload being constructed of commercially available parts, we consider this to be high-quality." Whether something is 'high-quality' cannot depend on how it comes about. There is a very recent paper by Cao et al. (2022) (https://acp.copernicus.org/articles/22/15379/2022/) showing that BRO can provide refractivity profiles with a standard deviation less than 1% between 10 and 15 km, in comparison to ERA5 (their Figure 8). The fact that it is not possible to provide this kind of quality with commercially available parts (at least not in this study) is an important message to get across. In any case, the abstract should objectively state the results of the study.

a. **Thank you for this comment. We have elected to remove the phrase "high-quality" from the sentence in question from the manuscript.**

2. L39: Perhaps missing an 'and' before "by fuel range of the aircraft."
   a. **Thank you for catching this error. It has been corrected in the manuscript.**

3. L124 (and many other places): I think the word 'delay' refers to the time delay of a signal, but for the phase the word 'excess' seems more appropriate (the signal is delayed in time, but there is an excess in the phase due to the neutral atmospheric influence). Please consider to remove 'delay' when talking about the phase.
   a. **Thank you for this comment. We have removed the word "delay" from references to excess phase in the manuscript.**

4. L142: Like the other reviewer, I find it relevant to discuss the impact of the POD uncertainty for BRO in step (b). In their answer (3a, bottom of page 9) to the other reviewer the authors write that "a brief discussion was added to reflect the concerns of the POD uncertainty.". But where? I don't see it.
   a. **Thank you for this comment. The discussion referenced in the prior round of revisions was probably insufficient. We have expanded the discussion of POD impacts on $N$-bias and moved it into section 4 of the manuscript: "… One potential cause is the difference in precise orbit determination (POD) solutions for BRO missions. Generally speaking, precise orbit determination (POD) solutions for LEO missions aim to have LEO velocity accuracies of 0.5 mm s$^{-1}$ or better, and LEO position accuracies of 10 cm or better. In contrast, BRO missions are generally capable of velocity accuracies of 30 mm s$^{-1}$ or better, and position accuracies of 5 cm or better. The larger POD velocity errors are due to difficult-to-model disturbances such as wind gusts and other aerodynamic factors. Xie et al., (2008) showed that the addition of simulated of 5 mm s$^{-1}$ random excess Doppler errors will not result in additional $N$-bias, but could possibly introduce less than 1% refractivity error near the receiver (~10 km) and less than 0.2% below ~ 6 km. In the case of our study, the larger BRO receiver positioning errors (if random) will not introduce significant $N$-bias (Fig. 9) for World View cases."**

5. L172-177: It is still unclear to me how many ERA5 profiles are used. The authors answer (15a) to my comment on this in the first review, and their revised manuscript, did not help. In their answer they write: "... median profile used to run the initial ROSAP simulation, run the initial Forward Abel Integrator, and determine the time series of refractivity at the receiver during the occultation.". So one profile for all this. But in the revised manuscript it says: "... one referencing refractivity profile .... at the zero-elevation ... used to compute the time series of refractivity at the receiver.... Furthermore ... we use a median refractivity profile ... surrounding the zero-elevation TP location for input into the initial ROSAP and FAI simulations." Thus, two profiles as I read it, one at zero elevation TP to determine the time series of refractivity at the receiver,

and another one (taking the median of grid points surrounding the zero-elevation TP) for ROSAP and FAI simulations. If their answer to me is correct, then the text needs to be updated to make it clear that it is the same profile used for these three tasks. I understand that a second (or third) profile at the 5 km TP location is used for comparisons because the lower TPs are drifting away from the zero-elevation TP location.

  a. **Thank you for this comment. We have rewritten Section 2.4 to better summarize the ERA5 refractivity profiles used during the retrieval process. The relevant portion now reads: "…. To best evaluate the quality of the individual retrieved BRO refractivity profiles, the final refractivity comparison uses the ERA5 profile at the 5 km TP location determined by ROSAP. Therefore, three separate refractivity profiles from ERA5 are used during the retrieval process, the zero-elevation angle refractivity profile, the median refractivity profile surrounding the zero-elevation angle location, and the 5 km TP location refractivity profile."**

6. Figure 5: I believe the dN (d for deci) in the x-axis label is the wrong unit. In the metric system 20 dm = 0.2 m. Thus 20 dN-units = 0.2 N-units. As mentioned in my first review, I think it should be deca N-units, so that 20 deca N-units = 200 N-units.

  a. **Thank you for catching this error. The reviewer is correct. After further investigation the abbreviation in the x-axis label should be "da" instead of "d." This change has been made to the figure and in the manuscript.**

7. Figure 5: The dashed lines need to be mentioned in the Figure caption. In their answer (16a) to my comment on this in the first review the authors write: "The dashed lines indicate the boundary layer height detected with the gradient method based on temperature, specific humidity, and refractivity (Nelson et al., 2021; Ao et al., 2012, 2008; Winning et al., 2017; Xie et al., 2006).". However, these papers are not referred to in the manuscript, where it just says (L189) "... weak gradients in specific humidity, and refractivity ...", which is not very understandable. I cannot see these weak gradients at 0.9 km in particular. Please mention the gradient method and the references in the paper. It is still not clear why there are two lines if they are based on temperature, specific humidity, and refractivity. Please make that clear in the paper. In the Figure caption it could say something like "The dashed lines at approximately 0.9 km indicate the PBLH (see text for details)."

  a. **Thank you for this comment. The gradient method for determining PBLH and selected references have been added to the manuscript. The various line styles and colors have also been added to the text. There are, indeed, three individual lines showing the PBLH. However, the PBLH identified from specific humidity and refractivity is exactly the same, so the lines overlap. Each line has a different style, and this is clarified in the text as well.**

8. L191-195: Please mention the sampling rate of the observations here or earlier in the paper. Are the simulations done at the same sampling rate? In their answer (19a), the

authors tell that the simulations has now been run through the same smoothing process as the observed excess phase data to remove high-frequency variability. This needs to be told in the paper.

    a. **Thank you for this comment. The explanation of the data processing and the retrieval process in Section 2.3 has been expanded to include further discussion of the sampling rates for the data and the smoothing algorithms applied.**

9. Figure 6: The ROSAP results look much better now. Not only is the high-frequency noise removed, but it also seems that the excess phase has come much closer to the calibrated observations than in the original figure (where the two curves crossed near 1000 sec). What is the reason for the latter? I don't suppose the smoothing can change the excess phase like that.

    a. **Thank you for this comment. In Figure 6, we have chosen to remove the raw, uncalibrated observations in addition to showing smoothed ROSAP excess phase and Doppler. Because the same smoothing algorithm is now applied to both the observed and simulated excess phase, the two datasets are much more similar to each other than they were previously.**

10. L199-202: I don't want to debate whether 141 and 700 is of the same order of magnitude as claimed by the authors in their answer (20a), and in the manuscript. To me the text is still misleading when it says: "The overall mean SNR from the GROOT receiver (141.79 V/V) is on the same order of magnitude (order of 100) as the mean SNR from the COSMIC-1 and SAC-C GNSS RO satellite missions (approximately 700 V/V ...)". I would rather say something like: "The overall mean SNR from the GROOT receiver is 141.79 V/V. This is relatively small compared to the mean SNR from the COSMIC-1 and SAC-C GNSS RO satellite missions, which is approximately 700 V/V." However, I think the COSMIC-1 and SAC-C SNRs are a bit smaller than 700 V/V due to defocusing at comparable tangent point altitudes (which could be relevant when you do such SNR comparisons). Please check. On the other hand, since the signal only passes through about half of the atmosphere in BRO relative to spaceborne RO, the defocusing effect in BRO will be smaller. This makes SNR comparison to spaceborne RO tricky. And how about the L2 signal? These discussions on SNR seems to focus only on L1, but the data quality also depends on the L2 signal. It should be mentioned in the paper that the discussion is about the L1 signal. It would be interesting to see also the SNR for the L2 signal in Fig. 6c. Please consider to add this.

    a. **Thank you for this comment. We have changed the discussion to focus on the L1 SNR to better reflect the reviewer's discussion on the differences between spaceborne and BRO SNR to the following: "The overall mean L1 SNR from the GROOT receiver (141.79 V V$^{-1}$) is smaller than the mean SNR from the COSMIC-1 and SAC-C GNSS RO satellite missions (approximately 700 V V$^{-1}$, Ao et al., 2009; Ho et al., 2020). Although the L1 SNR values from the Piksi receiver are approximately 5 times less than the values from spaceborne RO missions, considering the compact size of the Piksi receiver, such L1 SNR values are quite**

impressive. L2 frequency data was not as reliable as L1 frequency data and would often be interrupted or drop out earlier than L1. As such, the L2 data was only used for pre-processing data quality check (e.g., cycle slips), but not used for ionospheric correction." This has been added to the discussion in Section 2 for clarification as well.

b. **We agree with the reviewer regarding the complexity of the SNR comparison between airborne/balloon-borne and spaceborne RO due to the potential influence of the defocusing effect resulting from RO geometry differences. We have also added the following to the discussion in Section 3 highlighted below: "Although the L1 SNR values from the Piksi receiver are approximately 5 times less than the values from spaceborne RO missions, considering the compact size of the Piksi receiver, such L1 SNR values are quite impressive, which is partially attributed to the less defocusing effect than spaceborne RO as the Piksi receiver is in the atmosphere."**

11. Figure 7: How is the ROSAP bending angle calculated? From simulated excess Doppler? Or ray traced bending angle? The authors answer (22a) to this question did not help to clarify this. Please clarify in the paper.

a. **Thank you for this comment. The ROSAP bending angle is referring to the bending angle resulting from the ray-tracing simulation, so the total bending angle is calculated using accumulated changes in bending along each ray path at each impact parameter iteration. The text explaining the ROSAP simulation now reads: "which simulates the GNSS RO signal and calculates the associated excess phase and excess Doppler as well as the along-path accumulated bending angle at each impact parameter as it travels through a prescribed Earth's atmosphere (with either spherical or oblate Earth) by a given 1-dimensional atmospheric refractivity profile."**

12. L211-214: "Differences between the retrievals and the ROSAP simulation between ~6 and 8 km are most likely caused by differences between oblate and spherical Earth geometry assumptions. While we perform an Earth oblateness correction as part of the transmitter/receiver geometry processing, these effects may not be completely removed.". I very much doubt that to be the case. Why would the oblateness create something so distinct between 6 and 8 km, and then even after the oblateness correction? Is there a reason why you don't think this could be due to variations in the real atmosphere? Or could such variations be caused by yaw instability? Or ionospheric effects not adequately removed? Please revise the text unless you have substantial evidence (if so, please provide it) suggesting that this comes from the oblateness correction (you could just remove these new sentences if you don't have a likely explanation, and say that the variations are not understood). There are also differences around 11-12 km that should be discussed if these around 6-8 km are discussed.

a. **Thank you for this comment. We agree with the reviewer that the oblate correction in ROSAP is not the reason for the differences in the retrieved bending angles. On the contrary, ROSAP bending angle is expected to be closer**

to the "reality", if the input ERA5 profile represents the atmosphere well, which we believe to be true in a statistical sense. We hypothesize that the bending differences between the retrievals and the ROSAP simulation are more likely due to yaw rotation errors and their subsequent self-correction aboard the balloon platform. We see this more clearly with the feature between 10 and 11 km, where the retrieval bending increases only to relax back to values in line with both the ROSAP and Forward Abel simulations. These features also qualitatively coincide with changes in SNR (Fig. 6c) which is indicative of platform yaw rotation for a balloon platform. For the WVG26 case, the relative closeness of the retrievals to the Forward Abel simulation (assuming spherical symmetric atmosphere) opposed to the ROSAP (with oblateness correction) is likely by chance, as retrievals from other individual cases (not shown) do not necessarily match the Forward Abel better than ROSAP.**

   b. **As far as other potential reasons for the bending angle differences, although we did not carry out ionospheric corrections using L1/L2 linear combination due to the lower quality of L2 signal, we do expect the ionosphere effects are mostly corrected resulting from the bending angle differencing to compute partial bending angle. In addition, residual ionospheric effects should be more pronounced at higher altitude and is not seen in Figures 7 and 8. Therefore, it is unlikely that the variations are the result of ionospheric effects.**

13. L225-230: In their answer (24a) the authors explain that the median and MAD values referred to in the text are calculated over certain altitude regions: "As these are individual profiles, the median 'above 10 km' is a single value calculated from all data points above 10 km as well as within other regions.". But this is not explained in the revised text. Please do that if you prefer to discuss these median and MAD values. Later on (in Section 4) the median and MAD values are based on ensembles of profiles, which is quite different. Please make that clear. Alternatively (which I think would be much better), the discussion of median and MAD values based on the results in Fig. 8b in Section 3 could be omitted. It doesn't seem to add any useful information.

   a. **Thank you for this comment. We have removed the discussion of the median/MAD for the single profiles shown in Figure 8.**

14. L249-250: "The ZPM-1 refractivity differences from both retrieval methods show much more variability across all heights in both the median profile and the individual refractivity profiles." I disagree with this statement regarding the individual profiles. I see less variability in the grey lines in Fig. 10 than in Fig. 9. The document with tracked changes, and the authors answer (28a), mention "oscillations" that are not mentioned in the revised manuscript. Still, it is not clear what "oscillations" versus "variability" mean in this context. I think it would be better to acknowledge that there is less variability in the individual ZPM-1 results. The larger variability in the median seems to be a result of the lower number of profiles. In any case, there are so few profiles so that the results are likely not statistically significant.

a. **Thank you for this comment. We have rewritten the discussion of Figure 10 to indicate that the overall variability in the individual profiles is smaller than the World View campaign data. In the previous response to reviewer comments, we intended for the "oscillations" to refer to the 1000 m-scale features in Figure 10 where the *N*-bias increases over ~500 m then relaxes back to near its original value. Given the reviewer's comments we have elected to discuss them as "1000 m-scale vertical fluctuations in *N*-bias" rather than "oscillations."**

15. L258: "Reasons for errors … can come from … causes." Sentence does not make much sense. Instead, you could say: "Errors … can come from … causes".
    a. **Thank you for catching this grammar/syntax error. This has been updated in the manuscript to: "Errors in BRO refractivity retrievals can come from a variety of potential sources."**

16. L278: In this sentence the authors added 'and temporal': "The added benefit of using BRO platforms is the dense spatial and temporal sampling available due to the low platform velocities relative to the LEO-based RO satellites." It can be difficult to understand why the low platform velocities would increase the temporal sampling, but perhaps replacing the word 'available' with 'over targeted regions' in this sentence would more clearly describe what I think the authors mean.
    a. **Thank you for this comment. This change has been made in the manuscript.**

17. Figure A1: I still don't see any distinction between Piksi 1 and Piksi 2 in the text. In their answer (33a) the authors explain it nicely. Please consider to put that explanation in the Appendix.
    a. **Thank you for this comment. We have repurposed portions of our response from the previous review iteration that discusses the differences between Piksi 1 and Piksi 2 and placed it in the appendix as the following: "Balloon-borne RO can collect high density observations around the platform, particularly compared to spaceborne RO. Figure A1 shows a Sankey plot filtering visualization of the World View predicted occultations. Onboard the GROOT payload, we had duplicate Piksi receivers. Piksi 1 was programmed to log data from GPS, Galileo, Glonass, and Beidou, whereas Piksi 2 was only programmed to log data from GPS and Galileo. As such Piksi 2 was able to log more ROs from GPS. We believe the Piksi receivers and antennae were manufactured such that they were tuned to maximize GPS performance, and as such satellites from the GPS constellation consistently showed better performance than the other constellations. Of the original 680 predicted ROs from all GNSS constellations (originally discussed in section 2.1), a total of 485 were incomplete, and therefore not suitable for retrieval processing. The remaining 195 occultations then filter further by removing those from QZSS, Galileo, BeiDou, and GLONASS with low quality due to the frequency tuning inherent to the Piksi receiver. Of the 167 from the GPS system, only 8 good quality cases were ready to use immediately. Another 7 cases required additional pre-processing in the form of**

**cycle-slip corrections. The same process was also applied for cases observed during the ZPM-1 flight campaign.”**

**Comments from Reviewer #2:**

Manuscript Title: "GNSS Radio Occultation Soundings from Commercial Off-the-Shelf Receivers Onboard Balloon Platforms"

Manuscript Number: amt-2022-198

Date Reviewed: 19 DECEMBER 2022

Manuscript Evaluation: This work could potentially augment space-born ROs by better sampling the lower-to-middle troposphere. The authors have done an excellent job in addressing all my comments from the previous review cycle. I am not satisfied with the status of the current manuscript, and I recommend publication in its current form. However, I would like the authors (if they keep working on balloon ROs) to work on techniques to remove the SNR drops during BROs, as well as perform additional sensitivity studies to understand the influence of velocity accuracy on derived products.

**Thank you to the reviewer for their constructive suggestions and improvements to the manuscript.**